# Verified Relative Output Margins for Neural Network Twins

## Abstract

Given two neural network classifiers with the same input and output domains, our goal is to compare the two networks in relation to each other over an entire input region (e.g., within a vicinity of an input sample). Towards this, we introduce and quantify the Relative Output Margin (ROM) with which decisions are made. A larger output margin for a network w.r.t. another indicates that this network consistently makes a correct decision every time the other network does, and it does so in the entire input region. More importantly, as opposed to best-effort testing schemes, our framework is able to establish provably-correct (formally verified) bounds on ROM gains/losses over an entire input region. The proposed framework is relevant in the context of several application domains, e.g., for comparing a trained network and its corresponding compact (e.g., pruned, quantized, distilled) network. We evaluate our framework using the MNIST, CIFAR10, and two real-world medical datasets, to show its relevance.

## 1 Introduction

Quantitative comparison of neural networks, e.g., in terms of performance, is a fundamental concept in the Machine Learning (ML) domain. One common example is when a network is pruned, quantized, or distilled to run the compact networks on edge devices or smart sensors. In the medical domain, for instance, neural networks can enable implantable and wearable devices to detect cardiac arrhythmia (Sopic et al., 2018a) or epileptic seizures (Baghersalimi et al., 2024) in real time. However, due to their limited computing resources, such devices often adopt the compact networks corresponding to the original medical-grade networks. It is vital for the compact network to reliably detect cardiac abnormalities/seizures, as lack of reliable decisions can jeopardize patients' lives. Therefore, reasoning about the decisions made by the compact network w.r.t. to an original/reference network is vital for the safe deployment of the compact networks.

Traditionally, testing techniques have been adopted to quantitatively compare two neural networks. However, testing techniques often cannot cover all possible scenarios, hence do not provide any hard formal guarantees. This is required in the context of safety-critical applications, e.g., in the medical domain as discussed above. Therefore, here, we investigate providing hard correctness guarantees based on formal verification techniques.

In this work, we focus on neural network twins, i.e., two neural networks trained for the same learning/classification task, with the same input and output domains, but not the same weights and/or architectures. We define an input region as the region in a vicinity of a given input sample, e.g., captured by the absolute-value/Euclidean/maximum norm centered around the input sample. Given network twins and an input region, we investigate whether it is possible to prove that, *in the entire input region*, one network consistently makes a correct decision every time the other network does.

To this end, we consider the Output Margin (OM) of a network. Let us focus on binary classification for the simplicity of presentation. Given an input sample, the OM for one classifier corresponds to the magnitude of the change/perturbation in the output that leads to misclassification of the input sample. The larger the OM, the larger the adversarial perturbations needed to toggle the decision.

To quantitatively compare the neural network twins, we examine the OM of the two classifiers relative to each other, which we refer to as Relative Output Margin (ROM). Given the network twins and a

common input sample, ROM allows us to quantify the magnitude of the OM for one network relative to the other. Therefore, ROM captures whether the OM for one network is larger than the other.

We, then, extend the notion of ROM to Local Relative Output Margin (LROM), where we account for an entire input region (all perturbed inputs captured by the absolute-value/Euclidean/maximum norm centered around the input sample). That is, LROM not only enables us to reason about the given input samples, but also to reason about the entire region in the vicinity of the samples.

In this paper, we propose a framework to establish safe (provably-correct) bounds on LROM and formal verification guarantees on the decisions made by neural network twins in the entire input region. LROM enables us to formally prove that a network consistently makes a correct decision every time the other network does, and it does so in the entire input region. We evaluate our proposed framework extensively on several datasets to show its relevance, including two real-world medical applications for detection of cardiac arrhythmia or epileptic seizures. Our main contributions are summarized below:

- We formalize the notion of Local Relative Output Margins (LROMs) to quantitatively compare neural network twins in relation to each other in an entire input region.
- We propose a sound framework to derive (provably-correct) verified bounds on Local Relative Output Margins (LROMs) in an entire input region, to quantitatively compare the decisions of neural network twins.
- We conduct extensive experiments to compare the decisions made by pre-trained classifiers and their corresponding pruned, quantized, or knowledge-distilled counterparts on the MNIST dataset (LeCun, 1998), CIFAR10 dataset (Krizhevsky, 2009), CHB-MIT Scalp EEG database (Shoeb, 2010), and MIT-BIH Arrhythmia database (Goldberger et al., 2000).

## 2 LOCAL RELATIVE OUTPUT MARGINS (LROMs)

In this section, we formally describe Deep Neural Network (DNN) classifiers. Moreover, we introduce and formalize the notion of ROM and its extension to an entire input region, i.e., LROM.

### 2.1 DEEP NEURAL NETWORKS (DNNs)

In this work, we mainly consider DNN classifiers. A DNN classifier is a nonlinear function $\mathcal{N} : \mathbb{R}^{n_0^{\mathcal{N}}} \to \mathbb{R}^{n_N^{\mathcal{N}}}$ consisting of a sequence of $N$ layers followed by a softmax layer. Each layer is a linear transformation followed by a nonlinear activation function. Here, $n_k^{\mathcal{N}}$ is the number of neurons in the $k^{th}$ layer of network $\mathcal{N}$. Let $f_k^{\mathcal{N}}(\cdot) : \mathbb{R}^{n_{k-1}^{\mathcal{N}}} \to \mathbb{R}^{n_k^{\mathcal{N}}}$ be the function that derives values of the $k^{th}$ layer from the output of its preceding layer. The values of the $k^{th}$ layer, denoted by $\boldsymbol{x}^{(k)}$, are given by:

$$\boldsymbol{x}^{(k)} = f_k^{\mathcal{N}}(\boldsymbol{x}^{(k-1)}) = act_k^{\mathcal{N}}(\boldsymbol{W}^{(k)}\boldsymbol{x}^{(k-1)} + \boldsymbol{b}^{(k)}),$$

where $\boldsymbol{W}^{(k)}$ and $\boldsymbol{b}^{(k)}$ capture weights and biases of the $k^{th}$ layer, and $act_k^{\mathcal{N}}$ represents an activation function. The last layer uses softmax as the activation function to associate a probability to each class. For each class $c_i$ in the last layer $N + 1$, the softmax function value is: $\boldsymbol{x}_{c_i}^{(N+1)} = \sigma(\boldsymbol{x}^{(N)})_{c_i}$.

### 2.2 LOCAL RELATIVE OUTPUT MARGINS (LROMs)

We consider two DNNs $\mathcal{N}_1$ with $N_1 + 1$ layers with values $\boldsymbol{x}^{(0)}, \dots \boldsymbol{x}^{(N_1+1)}$ and $\mathcal{N}_2$ with $N_2 + 1$ layers with values $\boldsymbol{y}^{(0)}, \dots \boldsymbol{y}^{(N_2+1)}$. Suppose $n_0^{\mathcal{N}_1} = n_0^{\mathcal{N}_2}$ and $n_{N_1}^{\mathcal{N}_1} = n_{N_2}^{\mathcal{N}_2}$. Such networks are said to be *compatible/twins* as their inputs and outputs have the same dimensions.

Let us now introduce the notions of Output Margin (OM) and of Relative Output Margin (ROM).

**Definition 2.1.** Output Margin (OM) $\pi_{\boldsymbol{x}^{(0)}}^{\mathcal{N}_1}(c_i, c_j)$ of classes $(c_i, c_j)$ for DNN $\mathcal{N}_1$ and input $\boldsymbol{x}^{(0)}$ is the probabilities' ratio $\pi_{\boldsymbol{x}^{(0)}}^{\mathcal{N}_1}(c_i, c_j) = \frac{\sigma(\boldsymbol{x}^{(N_1)})_{c_i}}{\sigma(\boldsymbol{x}^{(N_1)})_{c_j}}$ of the outcome being $c_i$ by the one of being $c_j$.

Recall classifiers decide on the class with a maximum softmax value. Let us consider binary classification for the simplicity of presentation. Assuming the predicted class to be $c_i$, then

$\pi_{\boldsymbol{x}^{(0)}}^{\mathcal{N}_1}(c_i, c_j) = \frac{\sigma(\boldsymbol{x}^{(N_1)})_{c_i}}{\sigma(\boldsymbol{x}^{(N_1)})_{c_j}} \geq 1$. The closer the OM is to one, the more sensitive the decision is to per-

turbations, because even minor perturbations may toggle the decision, i.e., $\sigma(\boldsymbol{x}^{(N_1)})_{c_j} \geq \sigma(\boldsymbol{x}^{(N_1)})_{c_i}$.

**Definition 2.2.** Relative Output Margin (ROM) $\Pi_{\boldsymbol{x}^{(0)}}^{\mathcal{N}_1|\mathcal{N}_2}(c_i, c_j)$ of class pair $(c_i, c_j)$ for DNN $\mathcal{N}_1$
w.r.t. compatible DNN $\mathcal{N}_2$ and for common input $\boldsymbol{x}^{(0)} = \boldsymbol{y}^{(0)}$, is the quotient of OMs in $\mathcal{N}_1$ and $\mathcal{N}_2$:

$$\Pi_{\boldsymbol{x}^{(0)}}^{\mathcal{N}_1|\mathcal{N}_2}(c_i, c_j) = \frac{\pi_{\boldsymbol{x}^{(0)}}^{\mathcal{N}_1}(c_i, c_j)}{\pi_{\boldsymbol{y}^{(0)}}^{\mathcal{N}_2}(c_i, c_j)} = \frac{\sigma(\boldsymbol{x}^{(N_1)})_{c_i} \cdot \sigma(\boldsymbol{y}^{(N_2)})_{c_j}}{\sigma(\boldsymbol{x}^{(N_1)})_{c_j} \cdot \sigma(\boldsymbol{y}^{(N_2)})_{c_i}}.$$

We use $\Pi_{\boldsymbol{x}^{(0)}}^{\mathcal{N}_1|\mathcal{N}_2}(c_i, c_j)$ to compare the output margins (given a common input) between classes $c_i$
and $c_j$ in two compatible DNNs $\mathcal{N}_1$ and $\mathcal{N}_2$.

In this paper, our main goal is to establish bounds on ROM values in the entire input region,
e.g., in the vicinity of an input $\tilde{\boldsymbol{x}}^{(0)}$ or in a $\delta$-neighborhood of an input $\tilde{\boldsymbol{x}}^{(0)}$, defined as $\boldsymbol{D}_{\tilde{\boldsymbol{x}}^{(0)}}^{\delta} = \{\boldsymbol{x}^{(0)} \text{ s.t. } \|\boldsymbol{x}^{(0)} - \tilde{\boldsymbol{x}}^{(0)}\|_\infty \leq \delta\}$.

**Definition 2.3.** *Local Relative Output Margin (LROM)* of classes $(c_i, c_j)$ for DNN $\mathcal{N}_1$ w.r.t. compat-
ible DNN $\mathcal{N}_2$ in $\boldsymbol{D}_{\tilde{\boldsymbol{x}}^{(0)}}^{\delta}$ is the set $\left\{ \Pi_{\boldsymbol{x}^{(0)}}^{\mathcal{N}_1|\mathcal{N}_2}(c_i, c_j) \mid \boldsymbol{x}^{(0)} \in \boldsymbol{D}_{\tilde{\boldsymbol{x}}^{(0)}}^{\delta} \right\}$.

Note that, if $min \left\{ \Pi_{\boldsymbol{x}^{(0)}}^{\mathcal{N}_1|\mathcal{N}_2}(c_i, c_j) \mid \boldsymbol{x}^{(0)} \in \boldsymbol{D}_{\tilde{\boldsymbol{x}}^{(0)}}^{\delta} \right\} \geq 1$, then $\pi_{\boldsymbol{x}^{(0)}}^{\mathcal{N}_1}(c_i, c_j) \geq \pi_{\boldsymbol{x}^{(0)}}^{\mathcal{N}_2}(c_i, c_j)$, for all

$\boldsymbol{x}^{(0)}$ in the entire input region $\boldsymbol{D}_{\tilde{\boldsymbol{x}}^{(0)}}^{\delta}$. This, in turn, means that, in the entire input region $\boldsymbol{D}_{\tilde{\boldsymbol{x}}^{(0)}}^{\delta}$, $\mathcal{N}_1$
will make a correct decision every time $\mathcal{N}_2$ does.

## 3 METHOD

In this section, we introduce an optimization problem to bound LROMs for two compatible DNNs.
We also describe how we introduce and handle an over-approximation of the two networks in order
to soundly solve the optimization problem and derive a (provably-correct) verified bound.

### 3.1 THE LROM OPTIMIZATION PROBLEM

Assume two compatible DNNs $\mathcal{N}_1$ and $\mathcal{N}_2$ with respectively $N_1 + 1$ and $N_2 + 1$ layers, a common
input $\tilde{\boldsymbol{x}}^{(0)}$ in the domain $\boldsymbol{D}$ of $\mathcal{N}_1$ and $\mathcal{N}_2$, and a perturbation bound $\delta$. Our goal is to find, for any
class pair $(c_i, c_j)$, a tight lower bound for $min \left\{ \Pi_{\boldsymbol{x}^{(0)}}^{\mathcal{N}_1|\mathcal{N}_2}(c_i, c_j) \mid \boldsymbol{x}^{(0)} \in \boldsymbol{D}_{\tilde{\boldsymbol{x}}^{(0)}}^{\delta} \right\}$ and a tight upper
bound for $max \left\{ \Pi_{\boldsymbol{x}^{(0)}}^{\mathcal{N}_1|\mathcal{N}_2}(c_i, c_j) \mid \boldsymbol{x}^{(0)} \in \boldsymbol{D}_{\tilde{\boldsymbol{x}}^{(0)}}^{\delta} \right\}$.

Directly solving the above optimization problem involves the softmax function. Instead, we look into
$\ln \left( \Pi_{\boldsymbol{x}^{(0)}}^{\mathcal{N}_1|\mathcal{N}_2}(c_i, c_j) \right)$ and observe (Lemma A.1 in the appendix) it coincides with $(\boldsymbol{x}_{c_i}^{(N_1)} - \boldsymbol{x}_{c_j}^{(N_1)}) - (\boldsymbol{y}_{c_i}^{(N_2)} - \boldsymbol{y}_{c_j}^{(N_2)})$. Hence, we can characterize LROM bounds by reasoning on inputs to the softmax
layers (i.e., networks' logits). Therefore, our optimization objective is simplified to:

$$\ln \left( min \left\{ \Pi_{\boldsymbol{x}^{(0)}}^{\mathcal{N}_1|\mathcal{N}_2}(c_i, c_j) \mid \boldsymbol{x}^{(0)} \in \boldsymbol{D}_{\tilde{\boldsymbol{x}}^{(0)}}^{\delta} \right\} \right) = \min_{\boldsymbol{x}^{(0)} \in \boldsymbol{D}_{\tilde{\boldsymbol{x}}^{(0)}}^{\delta}} \left( (\boldsymbol{x}_{c_i}^{(N_1)} - \boldsymbol{x}_{c_j}^{(N_1)}) - (\boldsymbol{y}_{c_i}^{(N_2)} - \boldsymbol{y}_{c_j}^{(N_2)}) \right),$$

$$\ln \left( max \left\{ \Pi_{\boldsymbol{x}^{(0)}}^{\mathcal{N}_1|\mathcal{N}_2}(c_i, c_j) \mid \boldsymbol{x}^{(0)} \in \boldsymbol{D}_{\tilde{\boldsymbol{x}}^{(0)}}^{\delta} \right\} \right) = \max_{\boldsymbol{x}^{(0)} \in \boldsymbol{D}_{\tilde{\boldsymbol{x}}^{(0)}}^{\delta}} \left( (\boldsymbol{x}_{c_i}^{(N_1)} - \boldsymbol{x}_{c_j}^{(N_1)}) - (\boldsymbol{y}_{c_i}^{(N_2)} - \boldsymbol{y}_{c_j}^{(N_2)}) \right).$$

Building on the above, let $\mathcal{M}_{\tilde{\boldsymbol{x}}^{(0)},\delta}^{\mathcal{N}_1|\mathcal{N}_2}(c_i, c_j)$ be the value obtained as solution to the problem:

$$\mathcal{M}_{\tilde{\boldsymbol{x}}^{(0)},\delta}^{\mathcal{N}_1|\mathcal{N}_2}(c_i, c_j) = \min_{\boldsymbol{x}^{(0)}} \; (\boldsymbol{x}_{c_i}^{(N_1)} - \boldsymbol{x}_{c_j}^{(N_1)}) - (\boldsymbol{y}_{c_i}^{(N_2)} - \boldsymbol{y}_{c_j}^{(N_2)}), \tag{1}$$

$$\text{s.t.} \quad \boldsymbol{y}^{(0)} = \boldsymbol{x}^{(0)}, \quad \tilde{\boldsymbol{x}}^{(0)} \in \boldsymbol{D}, \tag{2}$$

$$\|\boldsymbol{x}^{(0)} - \tilde{\boldsymbol{x}}^{(0)}\|_\infty = \|\boldsymbol{y}^{(0)} - \tilde{\boldsymbol{x}}^{(0)}\|_\infty \leq \delta, \tag{3}$$

$$\boldsymbol{x}^{(k)} = f_k^{\mathcal{N}_1}(\boldsymbol{x}^{(k-1)}), \; \forall k \in \{1, \dots, N_1\}, \tag{4}$$

$$\boldsymbol{y}^{(l)} = f_l^{\mathcal{N}_2}(\boldsymbol{y}^{(l-1)}), \; \forall l \in \{1, \dots, N_2\}. \tag{5}$$

Equation (1) introduces the objective function used to capture the (logarithm of the) minimum ROM of the class pair $(c_i, c_j)$ for network $\mathcal{N}_1$ w.r.t. $\mathcal{N}_2$ in the input region $\boldsymbol{D}_{\tilde{\boldsymbol{x}}^{(0)}}^\delta$. Note that $\boldsymbol{x}_{c_i}^{(N_1)} - \boldsymbol{x}_{c_j}^{(N_1)}$ captures the difference between the logit values associated to classes $c_i$ and $c_j$ in network $\mathcal{N}_1$. Similarly, $\boldsymbol{y}_{c_i}^{(N_2)} - \boldsymbol{y}_{c_j}^{(N_2)}$ captures the difference between the logit values associated to the same classes in $\mathcal{N}_2$. The objective function is then to minimize the difference between these two quantities.

Let us consider Equations (2)–(5). Equation (2) enforces both that $\boldsymbol{y}^{(0)}$ (the perturbed input to network $\mathcal{N}_2$) equals $\boldsymbol{x}^{(0)}$ (the perturbed input to network $\mathcal{N}_1$), and that the original input $\tilde{\boldsymbol{x}}^{(0)}$ belongs to the dataset $\boldsymbol{D}$ of the two networks. Equation (3) enforces that the perturbed inputs $\boldsymbol{x}^{(0)}$ and $\boldsymbol{y}^{(0)}$ are in the $\delta$-neighborhood of $\tilde{\boldsymbol{x}}^{(0)}$. Equation (4) characterizes values of the first $N_1$ layers of network $\mathcal{N}_1$ as it relates the values of the $k^{th}$ layer (for $k$ in $\{1, \dots, N_1\}$) to those of its preceding layer, using the nonlinear function $f_k^{\mathcal{N}_1} : \mathbb{R}^{n_{k-1}^{\mathcal{N}_1}} \to \mathbb{R}^{n_k^{\mathcal{N}_1}}$. The same is applied to network $\mathcal{N}_2$ using the nonlinear functions $f_l^{\mathcal{N}_2} : \mathbb{R}^{n_{l-1}^{\mathcal{N}_2}} \to \mathbb{R}^{n_l^{\mathcal{N}_2}}$ for each layer $l$ as captured in Equation (5).

### 3.2 A Sound Over-Approximation of DNNs Behavior

Solving the above minimization problem is not trivial. Indeed, the activation functions result in nonlinear constraints for Equations (4) and (5). Here, we focus on Rectified Linear Unit (ReLU) functions, which are the most widely used activation functions in DNNs. Several recent approximation approaches have been proposed to tackle the nonlinearity of the activation functions in the context of verification problems for DNNs (Zhang et al., 2024; Baninajjar et al., 2023; Katz et al., 2019; Singh et al., 2019). To be able to capture ReLU, here, we consider existing relaxations (Ehlers, 2017; Baninajjar et al., 2023; Singh et al., 2019) to over-approximate the values computed at each layer using linear inequalities (described in Section A.2 in the appendix).

These over-approximations result in a relaxed optimization program that can be solved using Linear Programming (LP). The solution of the relaxed optimization problem is denoted by $\mathcal{R}_{\tilde{\boldsymbol{x}}^{(0)},\delta}^{\mathcal{N}_1|\mathcal{N}_2}(c_i, c_j)$. Because the relaxed optimization over-approximates the exact one in Equations (1)–(5) and that we are able to find the optimal solution to the LP relaxed formulation, any lower bound obtained for the relaxed problem is guaranteed to be smaller than a solution for the original minimization problem, i.e., $\mathcal{R}_{\tilde{\boldsymbol{x}}^{(0)},\delta}^{\mathcal{N}_1|\mathcal{N}_2}(c_i, c_j) \leq \mathcal{M}_{\tilde{\boldsymbol{x}}^{(0)},\delta}^{\mathcal{N}_1|\mathcal{N}_2}(c_i, c_j)$.

**Theorem 3.1.** *Let $(c_i, c_j)$ be a pair of classes of compatible DNNs $\mathcal{N}_1$ and $\mathcal{N}_2$. Assume a neighborhood $\boldsymbol{D}_{\tilde{\boldsymbol{x}}^{(0)}}^\delta$ and let $\mathcal{R}_{\tilde{\boldsymbol{x}}^{(0)},\delta}^{\mathcal{N}_1|\mathcal{N}_2}(c_i, c_j)$ (resp. $\mathcal{R}_{\tilde{\boldsymbol{x}}^{(0)},\delta}^{\mathcal{N}_2|\mathcal{N}_1}(c_i, c_j)$) be a solution to the relaxed minimization problem corresponding to LROM of $\mathcal{N}_1$ w.r.t. $\mathcal{N}_2$ (resp. $\mathcal{N}_2$ w.r.t. $\mathcal{N}_1$). Then:*

$$\mathcal{R}_{\tilde{\boldsymbol{x}}^{(0)},\delta}^{\mathcal{N}_1|\mathcal{N}_2}(c_i, c_j) \leq \mathcal{M}_{\tilde{\boldsymbol{x}}^{(0)},\delta}^{\mathcal{N}_1|\mathcal{N}_2}(c_i, c_j) = \ln\left(min\left\{\Pi_{\boldsymbol{x}^{(0)}}^{\mathcal{N}_1|\mathcal{N}_2}(c_i, c_j) \mid \boldsymbol{x}^{(0)} \in \boldsymbol{D}_{\tilde{\boldsymbol{x}}^{(0)}}^\delta\right\}\right)$$

$$\leq \ln\left(max\left\{\Pi_{\boldsymbol{x}^{(0)}}^{\mathcal{N}_1|\mathcal{N}_2}(c_i, c_j) \mid \boldsymbol{x}^{(0)} \in \boldsymbol{D}_{\tilde{\boldsymbol{x}}^{(0)}}^\delta\right\}\right)$$

$$= -\ln\left(min\left\{\Pi_{\boldsymbol{x}^{(0)}}^{\mathcal{N}_2|\mathcal{N}_1}(c_i, c_j) \mid \boldsymbol{x}^{(0)} \in \boldsymbol{D}_{\tilde{\boldsymbol{x}}^{(0)}}^\delta\right\}\right) = -\mathcal{M}_{\tilde{\boldsymbol{x}}^{(0)},\delta}^{\mathcal{N}_2|\mathcal{N}_1}(c_i, c_j) \leq -\mathcal{R}_{\tilde{\boldsymbol{x}}^{(0)},\delta}^{\mathcal{N}_2|\mathcal{N}_1}(c_i, c_j)$$

*Proof.* Proof sketch in appendix. $\square$

Theorem 3.1 not only provides a safe lower bound on LROM, i.e., $\mathcal{R}_{\tilde{\boldsymbol{x}}^{(0)},\delta}^{\mathcal{N}_1|\mathcal{N}_2}(c_i, c_j) \leq \mathcal{M}_{\tilde{\boldsymbol{x}}^{(0)},\delta}^{\mathcal{N}_1|\mathcal{N}_2}(c_i, c_j)$, but also a safe upper bound, i.e., $-\mathcal{R}_{\tilde{\boldsymbol{x}}^{(0)},\delta}^{\mathcal{N}_2|\mathcal{N}_1}(c_i, c_j) \geq \mathcal{M}_{\tilde{\boldsymbol{x}}^{(0)},\delta}^{\mathcal{N}_1|\mathcal{N}_2}(c_i, c_j)$.

An alternative approach to what we presented in this section is to reason based on independently-obtained ranges of Output Margins (OMs) for each network. However, such an approach results in an important loss of precision as it does not consider a common input to both networks, as we formalized in Theorem A.3 in the appendix and show empirically in Section 4.

## 4 EVALUATION

We evaluate our proposed framework and investigate ranges of LROMs for various datasets and DNNs.[1] All experiments are executed on a MacBook Pro equipped with an 8-core CPU and 32 GB of RAM using the Gurobi solver (Gurobi Optimization, LLC, 2023).

### 4.1 DATASETS

We use four different datasets for the evaluation of our framework, namely, the MNIST dataset (LeCun, 1998), CIFAR10 dataset (Krizhevsky, 2009), CHB-MIT Scalp EEG database (Shoeb, 2010) and MIT-BIH Arrhythmia database (Goldberger et al., 2000).

**MNIST dataset (LeCun, 1998)** contains grayscale handwritten digits, with each digit being depicted through a $28 \times 28$ pixel image. We consider the first 100 images of the test set, similar to (Ugare et al., 2022).

**CIFAR10 dataset (Krizhevsky, 2009)** comprises $32 \times 32$ colored images categorized into 10 different classes. In alignment with (Ugare et al., 2022), we focus on the first 100 images from the test set.

**CHB-MIT Scalp EEG database (Shoeb, 2010)** includes 23 individuals diagnosed with epileptic seizures. These recordings are sampled in the international 10–20 EEG system, and our focus is on F7-T7 and F8-T8 electrode pairs, commonly used in seizure detection (Sopic et al., 2018b).

**MIT-BIH Arrhythmia database (Goldberger et al., 2000)** involves 48 individuals with 2-channel ECG signals.To establish a classification problem, we consider a subset of 14 cardiac patients who demonstrated at least two different types of heartbeats.

### 4.2 NETWORKS

#### 4.2.1 ORIGINAL NETWORKS

For the MNIST and CIFAR10 datasets, we use fully-connected DNNs from (Ugare et al., 2022), which all have gone through robust training as outlined in (Chiang et al., 2020). They share the same structure, consisting of 7 dense layers with 200 neurons each. Patients in the CHB-MIT and MIT-BIH datasets have personalized convolutional DNNs. For each patient in the CHB-MIT dataset, the DNN has 2048 input neurons, two convolution layers followed by max-pooling layers with 3 and 5 filters, kernel sizes of 100 and 200, and a dense layer with 40 neurons. The accuracy ($\mu \pm \sigma$) is $85.7\% \pm 14.8\%$. For each patient in the MIT-BIH dataset, the DNN has an input layer with 320 neurons, a convolution layer with a 64-size kernel and 3 filters, and a dense layer with 40 neurons. The accuracy ($\mu \pm \sigma$) is $92.2\% \pm 9.1\%$.

#### 4.2.2 COMPACT NETWORKS

We explain the architecture and design of compact networks. Our experiments involve pruning, quantization, and knowledge distillation. These techniques are used to derive compact DNNs enabling energy-efficient inference on limited resources, and improving generalization and interoperability.

**Pruned Networks** are derived through a pruning procedure applied to DNNs, selectively nullifying certain weights and biases. Pruned networks maintain the architecture of their original counterparts. For the MNIST and CIFAR10 datasets, we use pruned networks generated by (Ugare et al., 2022) and (Baninajjar et al., 2024) through post-training pruning. Each pruned network generated by (Ugare et al., 2022) eliminates the smallest weights/biases in each layer, which is called Magnitude-Based Pruning (MBP), resulting in nine pruned networks with pruning rates ranging from 10% to 90%.

---

[1]The models, datasets, and code are included in the Supplementary Material.

(Baninajjar et al., 2024) produces Verification-friendly Neural Networks (VNNs) through the optimization of weights/biases, aiming to preserve their functionality while reducing the number of non-zero weights and biases. For the CHB-MIT and MIT-BIH datasets, we employ MBP pruning procedure where values below 10% of the maximum weight/bias are set to zero. In this context, accuracy ($\mu \pm \sigma$) is slightly reduced to $84.1\% \pm 17.2\%$ and $90.7\% \pm 10.9\%$ for the CHB-MIT and MIT-BIH datasets, respectively. Additionally, we utilize networks generated by (Baninajjar et al., 2024) with accuracies of $82.5\% \pm 9.6\%$ and $92.0\% \pm 9.1\%$.

**Quantized Networks** are obtained by quantization, where the precision of the networks' weights is reduced by converting them from 32-bit floating-point numbers to lower-precision representations. Quantized networks have the same architecture as their corresponding original networks. The DNNs of the MNIST and CIFAR10 datasets are presented by (Ugare et al., 2022) that are created by float16, int16, int8, and int4 post-training quantization. The same quantization is applied on individualized networks trained for CHB-MIT and MIT-BIH datasets. The accuracy ($\mu \pm \sigma$) of the int16 quantized networks is reported as $81.6\% \pm 14.8\%$ and $91.9\% \pm 10.1\%$ for the CHB-MIT and MIT-BIH datasets, respectively. The accuracy of other quantized networks can be found in the appendix.

**Distilled Networks** or student networks are compact networks trained using knowledge distillation to transfer information from larger teacher networks to mimic their behavior (Hinton et al., 2015). The architecture of distilled networks differs from the original ones. Furthermore, the temperature parameter affects the complexity of the distillation task, and we evaluate nine temperature values ranging from 1 to 9. As (Ugare et al., 2022) has not provided distilled networks for the MNIST and CIFAR10 datasets, we produce them using the methodology outlined in (Hinton et al., 2015). We consider DNNs featuring a single layer with 20 neurons for all the distilled networks. The same structure is employed to generate distilled networks for convolutional DNNs trained for CHB-MIT and MIT-BIH datasets. The accuracy ($\mu \pm \sigma$) of the distilled network with $T = 5$ is $71.6\% \pm 8.7\%$ and $89.7\% \pm 11.2\%$ for CHB-MIT and MIT-BIH datasets, respectively. Other accuracies can be found in the appendix.

## 4.3 RESULTS AND ANALYSIS

We conduct several experiments with our proposed method for establishing bounds on LROMs. We exclusively focus on correctly classified samples within each test set. We consider the widths and depths of the networks when defining perturbations. We use $\delta = 0.001$ and $\delta = 0.01$ for the MNIST and CIFAR datasets and experiment with several values for the CHB-MIH dataset ($\delta$ up to 0.002) and the MIT-BIH dataset ($\delta$ up to 0.4). We say LROM of $\mathcal{N}_1$ w.r.t. $\mathcal{N}_2$ is verified on a sample if we can establish, over the sample neighborhood, $\mathcal{R}^{\mathcal{N}_1 | \mathcal{N}_2}_{\tilde{\boldsymbol{x}}^{(0)}, \delta}(c, c_j) \geq 0$ for all pairs $(c, c_j)$, where $c$ is the correct class. This would mean that $\mathcal{M}^{\mathcal{N}_1 | \mathcal{N}_2}_{\tilde{\boldsymbol{x}}^{(0)}, \delta}(c, c_j) \geq 0$, because our method is sound, i.e., $\mathcal{R}^{\mathcal{N}_1 | \mathcal{N}_2}_{\tilde{\boldsymbol{x}}^{(0)}, \delta}(c_i, c_j) \leq \mathcal{M}^{\mathcal{N}_1 | \mathcal{N}_2}_{\tilde{\boldsymbol{x}}^{(0)}, \delta}(c_i, c_j)$. Here, we use zero as a threshold as it corresponds to checking increases or decreases of OM from one network to the other. However, our approach can easily accommodate other thresholds. We simply state that "$\mathcal{N}_1$ has a verified LROM sample" if LROM of $\mathcal{N}_1$ w.r.t. $\mathcal{N}_2$ is verified on the sample and $\mathcal{N}_2$ is clear from the context. Each time we verify LROM of $\mathcal{N}_1$ w.r.t. $\mathcal{N}_2$ for a sample, then the corresponding OM of any $(c, c_j)$, for correct class $c$, are guaranteed to be larger in $\mathcal{N}_1$ than the OM in $\mathcal{N}_2$. In addition, each time we show the upper bound of $\mathcal{N}_1$ w.r.t. $\mathcal{N}_2$ is negative (i.e., the lower bound of $\mathcal{N}_2$ w.r.t. $\mathcal{N}_1$ is positive) then the upper bound is indeed negative. In other words, if verified LROM of $\mathcal{N}_1$ w.r.t. $\mathcal{N}_2$, then the OMs for the correct class in $\mathcal{N}_2$ are indeed smaller than those in $\mathcal{N}_1$.

### 4.3.1 MNIST DATASET

Figures 1a– 1c describe the results of investigating LROMs for MNIST DNNs when $\delta = 0.001$. Figure 1a shows a noticeable rise in the percentage of verified LROM with increasing pruning proportions when investigating original networks w.r.t. pruned networks. There are two potential explanations for this phenomenon. First, the similarity between the original and less-pruned networks may result in no network having higher LROMs across the entire perturbation neighborhood. Second, our method may be capable of verifying LROMs of more samples in more-pruned networks, due to their sparsity. In addition, the last column of Figure 1a presents the results of investigating verified LROM of the VNN generated based on the original network using (Baninajjar et al., 2024). The

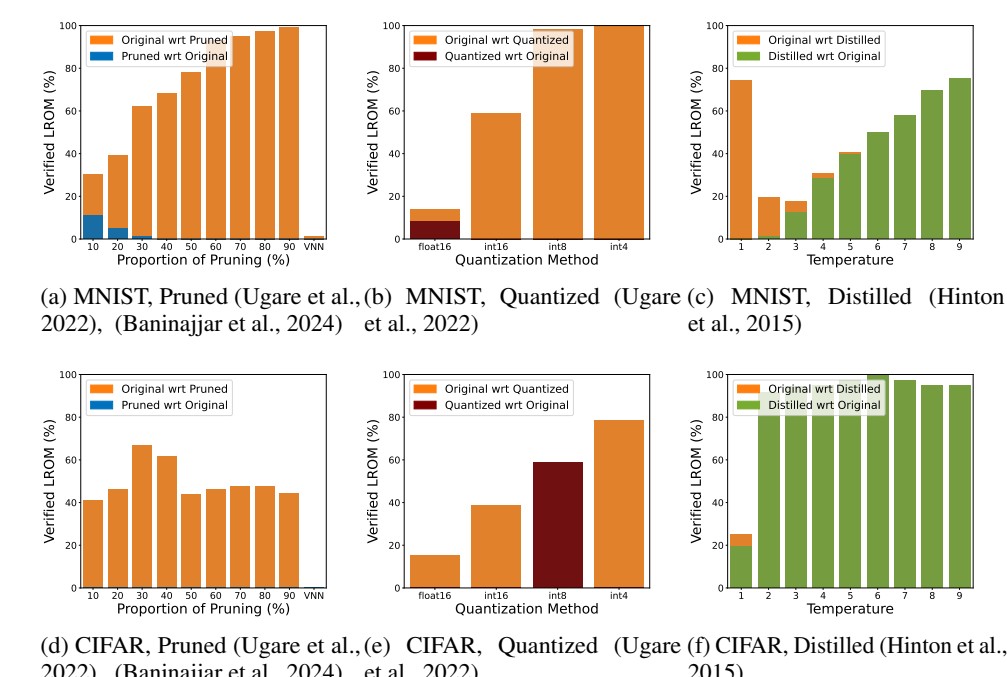

Figure 1: Stacked bar plots for verified LROM of DNNs trained on the MNIST and CIFAR10 datasets for different pruning methods, quantization precisions, and distillation temperatures for $\delta = 0.001$.

results indicate that the LROM of the VNN is comparable to that of the original network, since the verified LROM of both VNN and its corresponding original network is close to zero.

The percentages of verified LROM for quantized networks are depicted in Figures 1b, where the x-axis denotes quantization precision. Based on the results, original networks are more likely to have higher numbers of LROMs than quantized networks. However, Figure 1b shows the proportion of verified LROM for the quantized network with float16 precision is higher than the original one. Figure 1c represents percentages of verified LROMs of distilled networks w.r.t. original networks. The x-axis denotes the temperature of the distilled network, and the y-axis indicates the percentage of verified LROM. This figure shows the proportion of verified LROMs of distilled networks w.r.t. original networks increases as temperatures rise. The patterns of LROM exhibited by distilled networks set them apart from pruned and quantized networks, rendering them a favorable option for creating compact and energy-efficient networks.

**Processing Time.** The processing time of verifying LROM depends on the perturbation, i.e., the value of $\delta$, and the architecture of original and compact networks. In the case of the MNIST dataset, we exclusively take into account $\delta = 0.001$ and $\delta = 0.01$. The processing time ($\mu \pm \sigma$) is $15.0 \pm 0.7$ seconds when $\delta = 0.001$ and $18.3 \pm 5.2$ seconds when $\delta = 0.01$ for pruned and quantized networks. The processing time ($\mu \pm \sigma$) of distilled networks is $6.7 \pm 0.1$ and $6.9 \pm 0.2$ seconds for $\delta = 0.001$ and $\delta = 0.01$, respectively.

**Comparison with Independent Analysis.** Figures 2a– 2d demonstrate minimum and maximum LROMs for original networks w.r.t. the compact ones, using our method with joint analysis compared to the independent analysis. Due to the page limit, we present a pruned network with 50% pruning in Figure 2a, a VNN in Figure 2b, a quantized network with int16 precision in Figure 2c, and a distilled network with $T = 5$ in Figure 2d. The identity line divides the coordinate system into two sections; if a point lies above the identity line, it indicates that the point achieved a lower value using our method, and vice versa. These figures show that our method consistently achieves higher minimum LROMs, i.e., the corresponding values (triangles) are always below the identity line, compared to the independent analysis. Similarly, our method consistently achieves lower maximum LROMs, i.e., the corresponding values (circles) are always above the identity line. This indicates

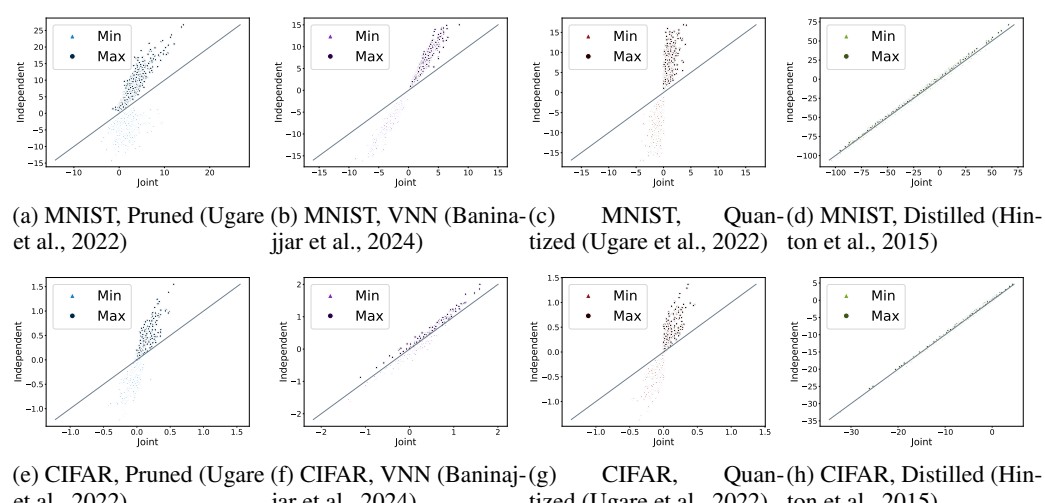

(a) MNIST, Pruned (Ugare et al., 2022)    (b) MNIST, VNN (Baninajjar et al., 2024)    (c) MNIST, Quantized (Ugare et al., 2022)    (d) MNIST, Distilled (Hinton et al., 2015)

(e) CIFAR, Pruned (Ugare et al., 2022)    (f) CIFAR, VNN (Baninajjar et al., 2024)    (g) CIFAR, Quantized (Ugare et al., 2022)    (h) CIFAR, Distilled (Hinton et al., 2015)

Figure 2: Minimum and maximum LROMs obtained by our method with joint analysis compared to independent analysis for original networks w.r.t. the compact ones when $\delta = 0.01$.

that our method reduces over-approximation in investigating LROMs, thereby finding minimum and maximum LROMs closer to the actual minimum and maximum LROMs. The further the points are from the identity line, the tighter the LROMs obtained by our framework. Figures 2a- 2c show the difference between independent and joint analysis is significant, as the points (both triangles and circles) significantly deviate from the identity line.

The results of verified LROM and comparison with independent analysis for convolutional DNNs trained on the MNIST dataset from (Ugare et al., 2022) are provided in the appendix.

**Adversarially-Trained Models.** As discussed earlier, our proposed framework is applicable to any two neural networks. In this section, we analyze three neural networks, two of which are adversarially-trained models that defend against adversarial attacks—specifically Projected Gradient Descent (PGD)—using different values of $\epsilon$. These networks share the same architecture with $6 \times 500$ neurons, as outlined in (Singh et al., 2019), and PGD-trained ones have $\epsilon$ values of 0.1 and 0.3, denoted as PGD1 and PGD3, respectively.

The investigation of LROM for all pairs of these three networks with $\delta = 0.001$ reveals that there is no sample such that the non-defended network exhibits higher OMs compared to its PGD-trained counterparts. Additionally, PGD1 and PGD3 consistently have more verified LROM than the non-defended network. These results become even more intriguing when compared to the outcomes of separately investigating the robustness of the neural networks using formal verification techniques. Considering $\delta = 0.001$, the certified accuracy of the non-defended, PGD1, and PGD3 networks is $100\%$ when each is evaluated individually using verification tools. Our framework highlights that, although robustness evaluations of neural networks might yield similar results, this does not imply that the networks behave identically. For instance, when considering a higher perturbation, such as $\delta = 0.01$, for individual robustness verification, the certified accuracy of the non-defended network drops to $89\%$, while the PGD-trained networks maintain a certified accuracy of $99\%$. This indicates that the results obtained by our framework accurately reflect the networks' behaviors.

Further investigations reveal that increasing $\delta$ to 0.04 leads to a more pronounced drop in the certified accuracy of PGD1 compared to PGD3, with PGD1's accuracy falling to $29\%$ while PGD3's remains at $87\%$. This is also reflected in our verified LROM results for $\delta = 0.001$, where PGD1 has $24\%$ verified LROM with respect to PGD3, compared to $38\%$ for PGD3 with respect to PGD1. Note that the remaining $38\%$ represents samples where neither of the PGD-trained networks consistently has higher OMs across all non-target classes.

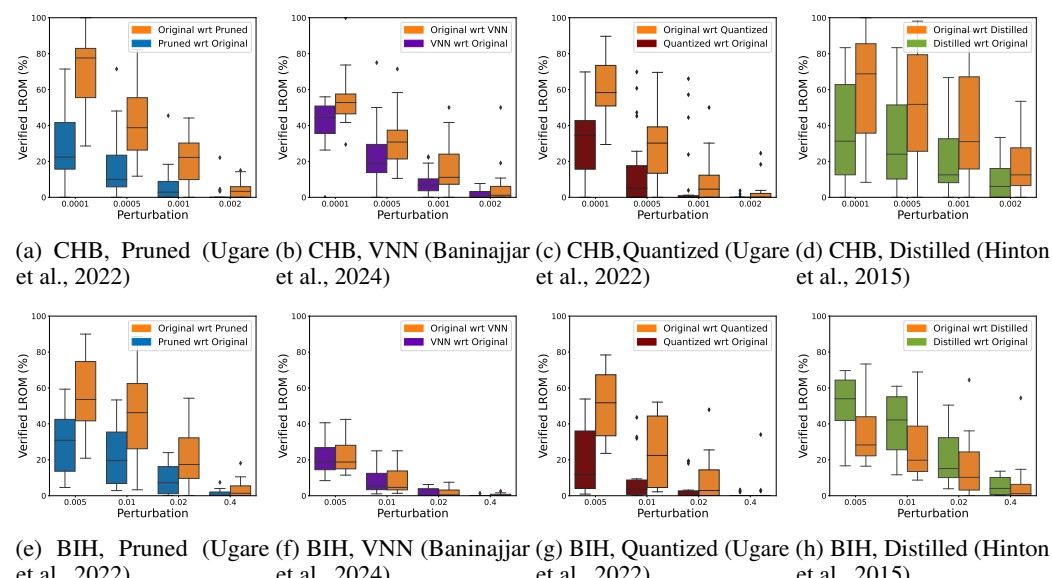

(a) CHB, Pruned (Ugare et al., 2022)

(b) CHB, VNN (Baninajjar et al., 2024)

(c) CHB, Quantized (Ugare et al., 2022)

(d) CHB, Distilled (Hinton et al., 2015)

(e) BIH, Pruned (Ugare et al., 2022)

(f) BIH, VNN (Baninajjar et al., 2024)

(g) BIH, Quantized (Ugare et al., 2022)

(h) BIH, Distilled (Hinton et al., 2015)

Figure 3: The box plots show verified LROM of original and compact convolutional DNNs trained for all patients of the CHB-MIT (Shoeb, 2010) and MIT-BIH (Goldberger et al., 2000) datasets.

### 4.3.2 CIFAR10 DATASET

Figures 1d– 1f describe the results of investigating LROMs for CIFAR10 DNNs when $\delta = 0.001$.

Although the general patterns in the results of the CIFAR10 DNNs are similar to those of the MNIST DNNs, a few differences are observed. In Figures 1e, the quantized network with the precision of int8 w.r.t. the original network has higher verified LROM. Moreover, in Figures 1f distilled networks consistently exhibit a higher number of verified LROM across different temperatures.

**Processing Time.** The processing time of CIFAR10 DNNs is higher than MNIST ones, as the number of parameters is higher due to the input size. The processing time ($\mu \pm \sigma$) is $33.1 \pm 2.1$ seconds when $\delta = 0.001$ and $43.9 \pm 8.1$ seconds when $\delta = 0.01$ for pruned and quantized networks. The processing time ($\mu \pm \sigma$) of distilled networks is $15.0 \pm 0.9$ and $18.1 \pm 2.6$ seconds for $\delta = 0.001$ and $\delta = 0.01$, respectively.

**Comparison with Independent Analysis.** Figures 2e– 2h demonstrate minimum and maximum LROMs for original networks w.r.t. the compact ones, using our joint analysis with our method vs. the independent analysis. Same as MNIST DNNs, we present a pruned network with 50% pruning in Figure 2e, a VNN in Figure 2f, a quantized network with int16 precision in Figure 2g, and a distilled network with $T = 5$ in Figure 2h, due to the page limit. The results from the CIFAR10 DNNs exhibit similarities to those of the MNIST DNNs, albeit with a less pronounced distinction between joint and independent analysis.

The results of verified LROM and comparison with independent analysis for convolutional DNNs trained on the CIFAR10 dataset from (Ugare et al., 2022) are provided in the appendix.

### 4.3.3 CHB-MIT DATASET

We explore the LROM of convolutional DNNs trained on the CHB-MIT dataset to categorize EEG signals of patients with epileptic seizures as captured in Figure 3a– 3d. Here, the x-axis shows different perturbation values applied to the input of a pair of original and compact networks. The general pattern of the LROMs of pruned (with both MBP and VNN methods), quantized, and distilled networks is that we could verify LROMs for more samples when the original networks were investigated w.r.t. the compact ones. Besides, the number of verified cases decreases by increasing perturbation. This can be caused by an actual decrease of LROM over a neighborhood, or by an exacerbated over-approximation as generated by the framework. Figure 3b shows that the average

verified LROM of VNNs is comparable to their original counterparts. Figure 3d compares original and distilled networks, concluding on more cases than pruned and quantized networks.

### 4.3.4 MIT-BIH DATASET

In this section, we assess the LROM of convolutional DNNs trained on the MIT-BIH dataset in categorizing ECG signals from patients with cardiac arrhythmia, as demonstrated in Figures 3e–3h. Similar to DNNs trained for the CHB-MIT dataset, the number of verified samples drops as perturbation increases, either due to reduced LROM across a range of perturbed inputs or increased over-approximation generated by the framework. The behavior of MBP pruned and quantized networks is also similar to CHB-MIT DNNs such that the LROM of original networks is higher than their corresponding pruned and quantized ones. However, the results of VNN pruned networks are slightly different whereas the verified LROMs are closer to their original counterparts. Moreover, distilled networks display a different pattern such that their verified LROM is higher than their corresponding original networks.

## 5 RELATED WORK

To investigate the impact of quantization on neural network, (Duncan et al., 2020) empirically show that quantization not only maintains robustness but can also enhance it, and generally, accuracy is preserved after the quantization process. Duncan defines robustness as the proportion of robust data points in the original model that are also robust in the quantized model. In contrast, our method also considers the margins with which the networks make their decisions, in addition to classification results. The results in (Duncan et al., 2020) indicate a reduction in number of misclassified inputs and a preservation of the robustness for given perturbations around them. We observe a decrease in the obtained margins. This suggests that models may remain robust, but they do so with reduced margins.

The studies by (LI et al., 2023) and (Jordao & Pedrini, 2021) show network pruning can empirically improve robustness of a trained network. However, our research indicates this is not always the case. (Wang et al., 2018) uses similar pruning methods as ours but applies two white box attacks including fast gradient sign method (FGSM) (Goodfellow et al., 2014) and projected gradient descent (PGD) (Madry et al., 2018). Their findings, which are consistent with ours, suggest setting small weights to zero can result in less robust networks. It appears that pruning solely for the purpose of reducing the number of parameters, without considering the overall accuracy of the network, can diminish its robustness. However, more deliberate pruning methods, such as stability-based pruning, might actually improve robustness.

## 6 CONCLUSIONS

In this work, we propose a framework to compare the two networks in relation to each other over an entire input region. Towards this, we introduce and quantify the Relative Output Margin (ROM) with which decisions are made. A larger output margin for a network w.r.t. another indicates that this network consistently makes a correct decision every time the other network does, and it does so in the entire input region. In addition, our framework allows establishing lower and upper bounds on the output margins, given an entire input region, to quantitatively compare neural networks twins. The proposed framework is relevant in the context of several application domains, e.g., for comparing a trained network and its corresponding compact (e.g., pruned, quantized, distilled) network. We evaluate our framework using the MNIST, CIFAR10, and two real-world medical datasets, to show its relevance.

**Limitations:** Built upon decades of progress in linear programming, our framework, while benefiting from the impressive polynomial time complexity of linear programming, is constrained by the problem sizes manageable by the current state-of-the-art linear programming toolboxes.

**Broader Impacts:** This paper presents work whose goal is to compare two DNNs in relation to each other. This is particularly important in the context of safety-critical applications, e.g., autonomous driving or medical applications, as we show in this paper using two medical case studies. This work enables us to quantify the safety of the decisions made by two DNNs in relation to each other.

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

## A APPENDIX

### A.1 PROOFS FROM SECTION 3

**Lemma A.1.** *Let $(c_i, c_j)$ be a pair of classes of compatible DNNs $\mathcal{N}_1$ and $\mathcal{N}_2$. Assume common input $\boldsymbol{x}^{(0)} = \boldsymbol{y}^{(0)}$. Suppose $\mathcal{N}_1$ has $N_1 + 1$ layers $\boldsymbol{x}^{(0)}, \dots \boldsymbol{x}^{(N_1+1)}$ and $\mathcal{N}_2$ has $N_2 + 1$ layers $\boldsymbol{y}^{(0)}, \dots \boldsymbol{y}^{(N_2+1)}$. Then:*

$$\ln\left(\Pi_{\boldsymbol{x}^{(0)}}^{\mathcal{N}_1|\mathcal{N}_2}(c_i, c_j)\right) = (\boldsymbol{x}_{c_i}^{(N_1)} - \boldsymbol{x}_{c_j}^{(N_1)}) - (\boldsymbol{y}_{c_i}^{(N_2)} - \boldsymbol{y}_{c_j}^{(N_2)})$$

*Proof.* By applying the $\ln$ function on the definition of $\Pi_{\boldsymbol{x}^{(0)}}^{\mathcal{N}_1|\mathcal{N}_2}(c_i, c_j)$:

$$\ln\left(\Pi_{\boldsymbol{x}^{(0)}}^{\mathcal{N}_1|\mathcal{N}_2}(c_i, c_j)\right) = \ln\left(\frac{\left(\sigma(\boldsymbol{x}^{(N_1)})\right)_{c_i} \cdot \left(\sigma(\boldsymbol{y}^{(N_2)})\right)_{c_j}}{\left(\sigma(\boldsymbol{x}^{(N_1)})\right)_{c_j} \cdot \left(\sigma(\boldsymbol{y}^{(N_2)})\right)_{c_i}}\right) = \ln\left(\frac{\frac{e^{\boldsymbol{x}_{c_i}^{(N_1)}}}{\sum_{u=1}^{n^{\mathcal{N}_1}} e^{\boldsymbol{x}_u^{(N_1)}}} \cdot \frac{e^{\boldsymbol{y}_{c_j}^{(N_2)}}}{\sum_{u=1}^{n^{\mathcal{N}_2}} e^{\boldsymbol{y}_u^{(N_2)}}}}{\frac{e^{\boldsymbol{x}_{c_j}^{(N_1)}}}{\sum_{u=1}^{n^{\mathcal{N}_1}} e^{\boldsymbol{x}_u^{(N_1)}}} \cdot \frac{e^{\boldsymbol{y}_{c_i}^{(N_2)}}}{\sum_{u=1}^{n^{\mathcal{N}_2}} e^{\boldsymbol{y}_u^{(N_2)}}}}\right)$$

$$= \ln\left(\frac{e^{\boldsymbol{x}_{c_i}^{(N_1)}} \cdot e^{\boldsymbol{y}_{c_j}^{(N_2)}}}{e^{\boldsymbol{x}_{c_j}^{(N_1)}} \cdot e^{\boldsymbol{y}_{c_i}^{(N_2)}}}\right) = \boldsymbol{x}_{c_i}^{(N_1)} + \boldsymbol{y}_{c_j}^{(N_2)} - (\boldsymbol{x}_{c_j}^{(N_1)} + \boldsymbol{y}_{c_i}^{(N_2)}) = \boldsymbol{x}_{c_i}^{(N_1)} - \boldsymbol{x}_{c_j}^{(N_1)} - (\boldsymbol{y}_{c_i}^{(N_2)} - \boldsymbol{y}_{c_j}^{(N_2)})$$

$\square$

**Corollary A.2.** *Let $(c_i, c_j)$ be a pair of classes of compatible DNNs $\mathcal{N}_1$ and $\mathcal{N}_2$ (with resp. $N_1 + 1$ and $N_2 + 1$ layers). Assume common input $\tilde{\boldsymbol{x}}^{(0)} = \tilde{\boldsymbol{y}}^{(0)}$ and perturbation $\delta$. For $\boldsymbol{x}^{(0)} = \boldsymbol{y}^{(0)}$ with $\boldsymbol{x}^{(0)} \in \boldsymbol{D}_{\tilde{\boldsymbol{x}}^{(0)}}^{\delta}$, let $\boldsymbol{x}^{(0)}, \dots \boldsymbol{x}^{(N_1+1)}$ be layers in $\mathcal{N}_1$ and $\boldsymbol{y}^{(0)}, \dots \boldsymbol{y}^{(N_2+1)}$ be layers in $\mathcal{N}_2$. Then:*

$$\ln\left(min\left\{\Pi_{\boldsymbol{x}^{(0)}}^{\mathcal{N}_1|\mathcal{N}_2}(c_i, c_j) \mid \boldsymbol{x}^{(0)} \in \boldsymbol{D}_{\tilde{\boldsymbol{x}}^{(0)}}^{\delta}\right\}\right) = min_{\boldsymbol{x}^{(0)} \in \boldsymbol{D}_{\tilde{\boldsymbol{x}}^{(0)}}^{\delta}}\left((\boldsymbol{x}_{c_i}^{(N_1)} - \boldsymbol{x}_{c_j}^{(N_1)}) - (\boldsymbol{y}_{c_i}^{(N_2)} - \boldsymbol{y}_{c_j}^{(N_2)})\right)$$

$$\ln\left(max\left\{\Pi_{\boldsymbol{x}^{(0)}}^{\mathcal{N}_1|\mathcal{N}_2}(c_i, c_j) \mid \boldsymbol{x}^{(0)} \in \boldsymbol{D}_{\tilde{\boldsymbol{x}}^{(0)}}^{\delta}\right\}\right) = max_{\boldsymbol{x}^{(0)} \in \boldsymbol{D}_{\tilde{\boldsymbol{x}}^{(0)}}^{\delta}}\left((\boldsymbol{x}_{c_i}^{(N_1)} - \boldsymbol{x}_{c_j}^{(N_1)}) - (\boldsymbol{y}_{c_i}^{(N_2)} - \boldsymbol{y}_{c_j}^{(N_2)})\right)$$

*Proof.* By applying the $\ln$ function on $min\left\{\Pi_{\boldsymbol{x}^{(0)}}^{\mathcal{N}_1|\mathcal{N}_2}(c_i, c_j) \mid \boldsymbol{x}^{(0)} \in \boldsymbol{D}_{\tilde{\boldsymbol{x}}^{(0)}}^{\delta}\right\}$ and using Lemma A.1:

$$\ln\left(min\left\{\Pi_{\boldsymbol{x}^{(0)}}^{\mathcal{N}_1|\mathcal{N}_2}(c_i, c_j) \mid \boldsymbol{x}^{(0)} \in \boldsymbol{D}_{\tilde{\boldsymbol{x}}^{(0)}}^{\delta}\right\}\right) = min\left\{\ln\left(\Pi_{\boldsymbol{x}^{(0)}}^{\mathcal{N}_1|\mathcal{N}_2}(c_i, c_j)\right) \mid \boldsymbol{x}^{(0)} \in \boldsymbol{D}_{\tilde{\boldsymbol{x}}^{(0)}}^{\delta}\right\}$$

$$= min\left\{\left((\boldsymbol{x}_{c_i}^{(N_1)} - \boldsymbol{x}_{c_j}^{(N_1)}) - (\boldsymbol{y}_{c_i}^{(N_2)} - \boldsymbol{y}_{c_j}^{(N_2)})\right) \mid \boldsymbol{x}^{(0)} \in \boldsymbol{D}_{\tilde{\boldsymbol{x}}^{(0)}}^{\delta}\right\}$$

$$= min_{\boldsymbol{x}^{(0)} \in \boldsymbol{D}_{\tilde{\boldsymbol{x}}^{(0)}}^{\delta}}\left((\boldsymbol{x}_{c_i}^{(N_1)} - \boldsymbol{x}_{c_j}^{(N_1)}) - (\boldsymbol{y}_{c_i}^{(N_2)} - \boldsymbol{y}_{c_j}^{(N_2)})\right)$$

$$\ln\left(max\left\{\Pi_{\boldsymbol{x}^{(0)}}^{\mathcal{N}_1|\mathcal{N}_2}(c_i, c_j) \mid \boldsymbol{x}^{(0)} \in \boldsymbol{D}_{\tilde{\boldsymbol{x}}^{(0)}}^{\delta}\right\}\right) = max\left\{\ln\left(\Pi_{\boldsymbol{x}^{(0)}}^{\mathcal{N}_1|\mathcal{N}_2}(c_i, c_j)\right) \mid \boldsymbol{x}^{(0)} \in \boldsymbol{D}_{\tilde{\boldsymbol{x}}^{(0)}}^{\delta}\right\}$$

$$= max\left\{\left((\boldsymbol{x}_{c_i}^{(N_1)} - \boldsymbol{x}_{c_j}^{(N_1)}) - (\boldsymbol{y}_{c_i}^{(N_2)} - \boldsymbol{y}_{c_j}^{(N_2)})\right) \mid \boldsymbol{x}^{(0)} \in \boldsymbol{D}_{\tilde{\boldsymbol{x}}^{(0)}}^{\delta}\right\}$$

$$= max_{\boldsymbol{x}^{(0)} \in \boldsymbol{D}_{\tilde{\boldsymbol{x}}^{(0)}}^{\delta}}\left((\boldsymbol{x}_{c_i}^{(N_1)} - \boldsymbol{x}_{c_j}^{(N_1)}) - (\boldsymbol{y}_{c_i}^{(N_2)} - \boldsymbol{y}_{c_j}^{(N_2)})\right)$$

$\square$

**Theorem A.3.** *Let $(c_i, c_j)$ be a pair of classes of compatible DNNs $\mathcal{N}_1$ and $\mathcal{N}_2$. Assume a neighborhood $\boldsymbol{D}_{\tilde{\boldsymbol{x}}^{(0)}}^{\delta}$ and let $\mathcal{R}_{\tilde{\boldsymbol{x}}^{(0)}, \delta}^{\mathcal{N}_1|\mathcal{N}_2}(c_i, c_j)$ (resp. $\mathcal{R}_{\tilde{\boldsymbol{x}}^{(0)}, \delta}^{\mathcal{N}_2|\mathcal{N}_1}(c_i, c_j)$) be a solution to the relaxed minimization problem corresponding to LROM of $\mathcal{N}_1$ w.r.t. $\mathcal{N}_2$ (resp. $\mathcal{N}_2$ w.r.t. $\mathcal{N}_1$). Then:*

$$\mathcal{R}_{\tilde{\boldsymbol{x}}^{(0)}, \delta}^{\mathcal{N}_1|\mathcal{N}_2}(c_i, c_j) \leq \ln\left(min\left\{\Pi_{\boldsymbol{x}^{(0)}}^{\mathcal{N}_1|\mathcal{N}_2}(c_i, c_j) \mid \boldsymbol{x}^{(0)} \in \boldsymbol{D}_{\tilde{\boldsymbol{x}}^{(0)}}^{\delta}\right\}\right)$$

*and*
$$\ln\left(max\left\{\Pi_{\boldsymbol{x}^{(0)}}^{\mathcal{N}_1|\mathcal{N}_2}(c_i, c_j) \mid \boldsymbol{x}^{(0)} \in \boldsymbol{D}_{\tilde{\boldsymbol{x}}^{(0)}}^{\delta}\right\}\right) \leq -\mathcal{R}_{\tilde{\boldsymbol{x}}^{(0)}, \delta}^{\mathcal{N}_2|\mathcal{N}_1}(c_i, c_j)$$

*Proof.* As mentioned in Section 3.2 any lower bound obtained for the relaxed problem is guaranteed to be smaller than a solution for the original minimization problem. Since $min\left\{\ln\left(\Pi_{\boldsymbol{x}^{(0)}}^{\mathcal{N}_1|\mathcal{N}_2}(c_i,c_j)\right) \mid \boldsymbol{x}^{(0)} \in \boldsymbol{D}_{\tilde{\boldsymbol{x}}^{(0)}}^{\delta}\right\} \geq \mathcal{R}_{\tilde{\boldsymbol{x}}^{(0)},\delta}^{\mathcal{N}_1|\mathcal{N}_2}(c_i,c_j)$, we get:

$$min\left\{\ln\left(\Pi_{\boldsymbol{x}^{(0)}}^{\mathcal{N}_1|\mathcal{N}_2}(c_i,c_j)\right) \mid \boldsymbol{x}^{(0)} \in \boldsymbol{D}_{\tilde{\boldsymbol{x}}^{(0)}}^{\delta}\right\}$$
$$= \ln\left(min\left\{\Pi_{\boldsymbol{x}^{(0)}}^{\mathcal{N}_1|\mathcal{N}_2}(c_i,c_j) \mid \boldsymbol{x}^{(0)} \in \boldsymbol{D}_{\tilde{\boldsymbol{x}}^{(0)}}^{\delta}\right\}\right) \geq \mathcal{R}_{\tilde{\boldsymbol{x}}^{(0)},\delta}^{\mathcal{N}_1|\mathcal{N}_2}(c_i,c_j)$$

And similarly in a symmetric manner:

$$min\left\{\ln\left(\Pi_{\boldsymbol{x}^{(0)}}^{\mathcal{N}_2|\mathcal{N}_1}(c_i,c_j)\right) \mid \boldsymbol{x}^{(0)} \in \boldsymbol{D}_{\tilde{\boldsymbol{x}}^{(0)}}^{\delta}\right\}$$
$$= min\left\{\ln\left(\frac{\sigma(\boldsymbol{x}^{(\mathcal{N}_2)})_{c_i}\cdot\sigma(\boldsymbol{y}^{(\mathcal{N}_1)})_{c_j}}{\sigma(\boldsymbol{x}^{(\mathcal{N}_2)})_{c_j}\cdot\sigma(\boldsymbol{y}^{(\mathcal{N}_1)})_{c_i}}\right) \mid \boldsymbol{x}^{(0)} \in \boldsymbol{D}_{\tilde{\boldsymbol{x}}^{(0)}}^{\delta}\right\} = min\left\{-\ln\left(\frac{\sigma(\boldsymbol{y}^{(\mathcal{N}_1)})_{c_i}\cdot\sigma(\boldsymbol{x}^{(\mathcal{N}_2)})_{c_j}}{\sigma(\boldsymbol{y}^{(\mathcal{N}_1)})_{c_j}\cdot\sigma(\boldsymbol{x}^{(\mathcal{N}_2)})_{c_i}}\right) \mid \boldsymbol{x}^{(0)} \in \boldsymbol{D}_{\tilde{\boldsymbol{x}}^{(0)}}^{\delta}\right\}$$
$$= -max\left\{\ln\left(\frac{\sigma(\boldsymbol{y}^{(\mathcal{N}_1)})_{c_i}\cdot\sigma(\boldsymbol{x}^{(\mathcal{N}_2)})_{c_j}}{\sigma(\boldsymbol{y}^{(\mathcal{N}_1)})_{c_j}\cdot\sigma(\boldsymbol{x}^{(\mathcal{N}_2)})_{c_i}}\right) \mid \boldsymbol{x}^{(0)} \in \boldsymbol{D}_{\tilde{\boldsymbol{x}}^{(0)}}^{\delta}\right\} = -max\left\{\ln\left(\Pi_{\boldsymbol{x}^{(0)}}^{\mathcal{N}_1|\mathcal{N}_2}(c_i,c_j)\right) \mid \boldsymbol{x}^{(0)} \in \boldsymbol{D}_{\tilde{\boldsymbol{x}}^{(0)}}^{\delta}\right\}$$
$$= -\ln\left(max\left\{\Pi_{\boldsymbol{x}^{(0)}}^{\mathcal{N}_1|\mathcal{N}_2}(c_i,c_j) \mid \boldsymbol{x}^{(0)} \in \boldsymbol{D}_{\tilde{\boldsymbol{x}}^{(0)}}^{\delta}\right\}\right) \geq \mathcal{R}_{\tilde{\boldsymbol{x}}^{(0)},\delta}^{\mathcal{N}_2|\mathcal{N}_1}(c_i,c_j)$$

Then we can deduce:

$$\mathcal{R}_{\tilde{\boldsymbol{x}}^{(0)},\delta}^{\mathcal{N}_1|\mathcal{N}_2}(c_i,c_j) \quad \leq \quad \ln\left(min\left\{\Pi_{\boldsymbol{x}^{(0)}}^{\mathcal{N}_1|\mathcal{N}_2}(c_i,c_j) \mid \boldsymbol{x}^{(0)} \in \boldsymbol{D}_{\tilde{\boldsymbol{x}}^{(0)}}^{\delta}\right\}\right)$$

$$\text{and} \qquad \ln\left(max\left\{\Pi_{\boldsymbol{x}^{(0)}}^{\mathcal{N}_1|\mathcal{N}_2}(c_i,c_j) \mid \boldsymbol{x}^{(0)} \in \boldsymbol{D}_{\tilde{\boldsymbol{x}}^{(0)}}^{\delta}\right\}\right) \quad \leq \quad -\mathcal{R}_{\tilde{\boldsymbol{x}}^{(0)},\delta}^{\mathcal{N}_2|\mathcal{N}_1}(c_i,c_j)$$

$\square$

## A.2 A Sound Over-Approximation of DNNs Behavior

Solving the original minimization problem is not trivial. Indeed, the activation functions result in nonlinear constraints for Equations (4) and (5). Our analysis targets ReLU layers, it can be generalized to accommodate any nonlinear activation function that can be represented in a piece-wise linear form (Ehlers, 2017). ReLU functions are the most widely used activation functions in DNNs. Recall the last layer is a softmax layer, but we are only interested in the possible values of its inputs. We explain in the following how to over-approximate the values computed at each layer using linear inequalities. The goal is to make possible the computation of a tight lower bound for the minimization problem from Section 3.1.

A ReLU compounds two linear segments, resulting in a piece-wise linear function. Consider $\hat{x}_i^{(k)} = \boldsymbol{W}_{i,:}^{(k)}\boldsymbol{x}^{(k-1)} + b_i^{(k)}$, the value of the $i^{th}$ neuron in the $k^{th}$ layer before applying the activation function. The output $x_i^{(k)}$ of the ReLU of $\hat{x}_i^{(k)}$ is $\hat{x}_i^{(k)}$ if $\hat{x}_i^{(k)} \geq 0$ and 0 otherwise. When considering a $\delta$-neighborhood as inputs, each neuron $\hat{x}_i^{(k)}$ gets lower and upper bounds, denoted as $\underline{\hat{x}}_i^{(k)}$ and $\overline{\hat{x}}_i^{(k)}$, respectively. Applying ReLU to each neuron $\hat{x}_i^{(k)}$ results in the neuron being always active when both lower and upper bounds are positive (i.e., ReLU coincides with the identity relation), and always inactive when both are negative (i.e., ReLU coincides with zero). There is a third situation where lower and upper bounds have different signs. To adapt ReLU to our optimization framework, we consider as in (Ehlers, 2017) the minimum convex area bounded by $\underline{\hat{x}}_i^{(k)}$ and $\overline{\hat{x}}_i^{(k)}$. The convex is given by the three inequalities:

$$x_i^{(k)} \leq \overline{\hat{x}}_i^{(k)} \cdot \frac{\hat{x}_i^{(k)} - \underline{\hat{x}}_i^{(k)}}{\overline{\hat{x}}_i^{(k)} - \underline{\hat{x}}_i^{(k)}}, \quad x_i^{(k)} \geq \hat{x}_i^{(k)}, \quad x_i^{(k)} \geq 0.$$

Lower and upper bounds of each neuron can be calculated by propagating through the network, starting from the input layer w.r.t. the perturbation $\delta$. In fact, our proposed framework can manage various layers including, but not limited to, convolution, zero-padding, max-pooling, permute, and

flattening layers. For instance, a max-pooling layer with a pool size of $p_k$ can be approximated with $p_k + 1$ inequalities as follows. Let $J = \{(i-1)p_k + 1, \ldots, ip_k\}$, use:

$$x_i^{(k)} \geq x_j^{(k-1)}, \forall j \in J,$$

$$\sum_{j \in J} x_j^{(k-1)} \geq x_i^{(k)} + \sum_{j \in J} \underline{x}_j^{(k-1)} - \max_{j \in J} \underline{x}_j^{(k-1)}.$$

Other nonlinear layers used in Equations (4) and (5) can also be over-approximated using linear inequalities.

## A.3 Joint vs Independent Analysis

We abuse notation and write $\mathcal{M}_{\tilde{\boldsymbol{x}}^{(0)}, \delta}^{\mathcal{N}_1 | \mathbf{0}}(c_i, c_j)$ to mean the value of the objective function in Equation (1) (of the original optimization problem in Section 3) when choosing a constant second network $\mathcal{N}_2$ that assigns equal probabilities to each outcome. This corresponds to computing minimum Output Margins (OMs) for $\mathcal{N}_1$ on its own. Our original optimization problem and its linear relaxation compute ROMs' bounds for a given input that is common to both networks and that is ranging over the considered neighborhood. This can be simplified by independently computing ranges of Output Margins (OMs) for each network and by combining the results. This would result in sound approximations of ROMs. However, this decoupled approach results in a loss of precision as it does not consider a common input to both networks. This is formalized by the theorem below, and is witness by our experiments where we evaluate the corresponding loss in precision. We report on the experiments in Section 4.

**Theorem A.4.** $\mathcal{M}_{\tilde{\boldsymbol{x}}^{(0)}, \delta}^{\mathcal{N}_1 | \mathbf{0}}(c_i, c_j) + \mathcal{M}_{\tilde{\boldsymbol{x}}^{(0)}, \delta}^{\mathbf{0} | \mathcal{N}_2}(c_i, c_j) \leq \mathcal{M}_{\tilde{\boldsymbol{x}}^{(0)}, \delta}^{\mathcal{N}_1 | \mathcal{N}_2}(c_i, c_j)$

*Proof.* Recall:

- $\mathcal{M}_{\tilde{\boldsymbol{x}}^{(2)}, \delta}^{\mathcal{N}_1 | \mathcal{N}_2}(c_i, c_j) = min\left\{\left((\boldsymbol{x}_{c_i}^{(N_1)} - \boldsymbol{x}_{c_j}^{(N_1)}) - (\boldsymbol{y}_{c_i}^{(N_2)} - \boldsymbol{y}_{c_j}^{(N_2)})\right) \mid \boldsymbol{x}^{(0)} \in \boldsymbol{D}_{\tilde{\boldsymbol{x}}^{(0)}}^{\delta}\right\}$

- $\mathcal{M}_{\tilde{\boldsymbol{x}}^{(0)}, \delta}^{\mathcal{N}_1 | \mathbf{0}}(c_i, c_j) = min\left\{\left((\boldsymbol{x}_{c_i}^{(N_1)} - \boldsymbol{x}_{c_j}^{(N_1)})\right) \mid \boldsymbol{x}^{(0)} \in \boldsymbol{D}_{\tilde{\boldsymbol{x}}^{(0)}}^{\delta}\right\}$

- $\mathcal{M}_{\tilde{\boldsymbol{x}}^{(0)}, \delta}^{\mathbf{0} | \mathcal{N}_2}(c_i, c_j) = min\left\{\left(-(\boldsymbol{y}_{c_i}^{(N_2)} - \boldsymbol{y}_{c_j}^{(N_2)})\right) \mid \boldsymbol{y}^{(0)} \in \boldsymbol{D}_{\tilde{\boldsymbol{x}}^{(0)}}^{\delta}\right\}$

Let $\boldsymbol{x}_{c_i}^{(N_1)}, \boldsymbol{x}_{c_j}^{(N_1)}, \boldsymbol{y}_{c_i}^{(N_2)}, \boldsymbol{y}_{c_j}^{(N_2)}$ be the logits values obtained in the solution $\mathcal{M}_{\tilde{\boldsymbol{x}}^{(2)}, \delta}^{\mathcal{N}_1 | \mathcal{N}_2}(c_i, c_j)$.

Let $\boldsymbol{x}_{c_i}^{(N_1)'}, \boldsymbol{x}_{c_j}^{(N_1)'}$ be the logits values obtained in the solution $\mathcal{M}_{\tilde{\boldsymbol{x}}^{(2)}, \delta}^{\mathcal{N}_1 | \mathbf{0}}(c_i, c_j)$.

Let $\boldsymbol{y}_{c_i}^{(N_2)'}, \boldsymbol{y}_{c_j}^{(N_2)'}$ be the logits values obtained in the solution $\mathcal{M}_{\tilde{\boldsymbol{x}}^{(2)}, \delta}^{\mathbf{0} | \mathcal{N}_2}(c_i, c_j)$.

By definitions:

- $(\boldsymbol{x}_{c_i}^{(N_1)'} - \boldsymbol{x}_{c_j}^{(N_1)'}) \leq (\boldsymbol{x}_{c_i}^{(N_1)} - \boldsymbol{x}_{c_j}^{(N_1)})$

- $-(\boldsymbol{y}_{c_i}^{(N_2)'} - \boldsymbol{y}_{c_j}^{(N_2)'}) \leq -(\boldsymbol{y}_{c_i}^{(N_2)} - \boldsymbol{y}_{c_j}^{(N_2)})$

Hence:

$$(\boldsymbol{x}_{c_i}^{(N_1)'} - \boldsymbol{x}_{c_j}^{(N_1)'}) - (\boldsymbol{y}_{c_i}^{(N_2)'} - \boldsymbol{y}_{c_j}^{(N_2)'}) \leq (\boldsymbol{x}_{c_i}^{(N_1)} - \boldsymbol{x}_{c_j}^{(N_1)}) - (\boldsymbol{y}_{c_i}^{(N_2)} - \boldsymbol{y}_{c_j}^{(N_2)})$$

and

$$\mathcal{M}_{\tilde{\boldsymbol{x}}^{(0)}, \delta}^{\mathcal{N}_1 | \mathbf{0}}(c_i, c_j) + \mathcal{M}_{\tilde{\boldsymbol{x}}^{(0)}, \delta}^{\mathbf{0} | \mathcal{N}_2}(c_i, c_j) \leq \mathcal{M}_{\tilde{\boldsymbol{x}}^{(0)}, \delta}^{\mathcal{N}_1 | \mathcal{N}_2}(c_i, c_j)$$

$\square$

### A.4 ADDITIONAL TABLES AND FIGURES FOR SECTION 4

In this section, we provide supplementary experiments.

#### A.4.1 MNIST AND CIFAR10 DATASETS

Together with the fully-connected DNNs, there are convolutional DNNs trained on the MNIST and CIFAR10 datasets provided by (Ugare et al., 2022). The convolutional DNN trained on the MNIST dataset includes two convolution layers, each preceded by a zero-padding layer, followed by five dense layers, each comprising 256 neurons. The convolutional DNN trained on the CIFAR10 dataset has two additional pairs of convolution and zero-padding layers compared to the MNIST' convolutional DNN. Figure 4 presents the stacked bar plots of verified LROMs obtained using our method on the convolutional DNNs when $\delta = 0.001$. Figure 5 demonstrates the minimum and maximum values of LROMs achieved by our method with joint analysis, compared to the values obtained by independent analysis of the networks when $\delta = 0.01$.

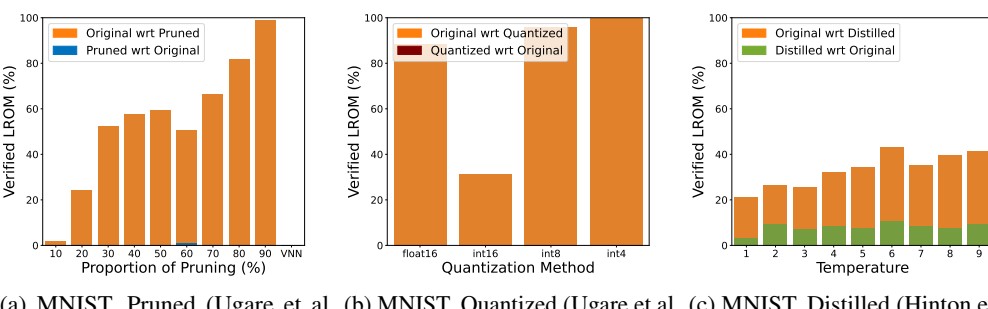

(a) MNIST, Pruned (Ugare et al., 2022), (Baninajjar et al., 2024)

(b) MNIST, Quantized (Ugare et al., 2022)

(c) MNIST, Distilled (Hinton et al., 2015)

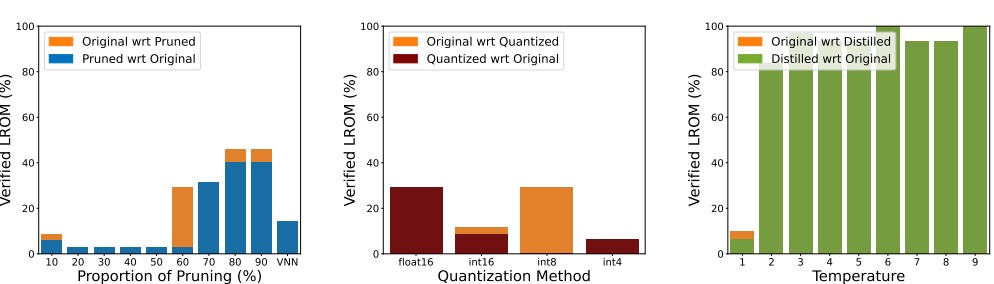

(d) CIFAR, Pruned (Ugare et al., 2022), (Baninajjar et al., 2024)

(e) CIFAR, Quantized (Ugare et al., 2022)

(f) CIFAR, Distilled (Hinton et al., 2015)

Figure 4: Stacked bar plots for verified LROM of convolutional DNNs trained on the MNIST and CIFAR10 datasets when $\delta = 0.001$.

#### A.4.2 CHB-MIT AND MIT-BIH DATASETS

Here, we provide the accuracy ($\mu \pm \sigma$) of quantized and distilled networks generated for CHB-MIT and MIT-BIH datasets in Tables 1 and 2, respectively.

Table 1: Accuracy of quantized networks trained on CHB-MIT and MIT-BIH datasets.

|         | float16          | int8             | int4             |
|---------|------------------|------------------|------------------|
| CHB-MIT | $85.7 \pm 14.8$  | $80.9 \pm 15.0$  | $85.1 \pm 15.1$  |
| MIT-BIH | $92.2 \pm 10.1$  | $91.8 \pm 10.1$  | $90.9 \pm 10.7$  |

Moreover, we assess the verified LROMs of quantized and distilled networks derived from convolutional DNNs trained on the CHB-MIT and MIT-BIH datasets, shown in Figure 6 and 7. These networks have different precision/temperature compared to those mentioned in Section 4.

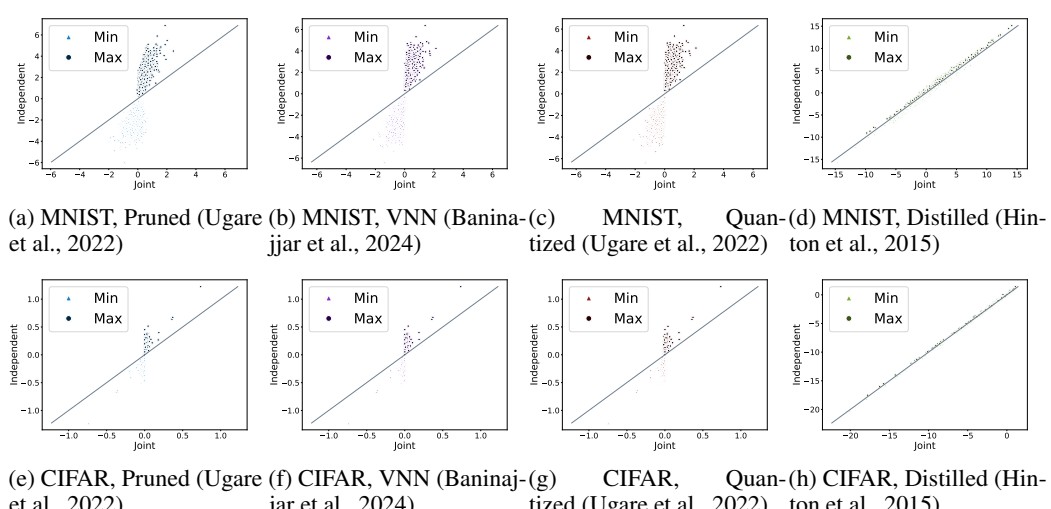

(a) MNIST, Pruned (Ugare et al., 2022) (b) MNIST, VNN (Baninajjar et al., 2024) (c) MNIST, Quantized (Ugare et al., 2022) (d) MNIST, Distilled (Hinton et al., 2015)

(e) CIFAR, Pruned (Ugare et al., 2022) (f) CIFAR, VNN (Baninajjar et al., 2024) (g) CIFAR, Quantized (Ugare et al., 2022) (h) CIFAR, Distilled (Hinton et al., 2015)

Figure 5: Minimum and maximum LROMs obtained by our method with joint analysis compared to independent analysis for original networks w.r.t. the compact ones when $\delta = 0.01$.

Table 2: Accuracy of distilled networks trained on CHB-MIT and MIT-BIH datasets.

|  | T1 | T2 | T3 | T4 | T6 | T7 | T8 | T9 |
|---|---|---|---|---|---|---|---|---|
| CHB-MIT | $73.4 \pm 9.5$ | $72.5 \pm 11.2$ | $73.0 \pm 9.4$ | $71.0 \pm 9.0$ | $71.5 \pm 9.6$ | $73.5 \pm 9.7$ | $73.5 \pm 8.6$ | $71.5 \pm 9.5$ |
| MIT-BIH | $91.9 \pm 6.7$ | $90.6 \pm 9.2$ | $90.0 \pm 11.2$ | $89.1 \pm 12.7$ | $90.3 \pm 10.2$ | $90.6 \pm 10.3$ | $90.6 \pm 10.2$ | $90.2 \pm 10.0$ |

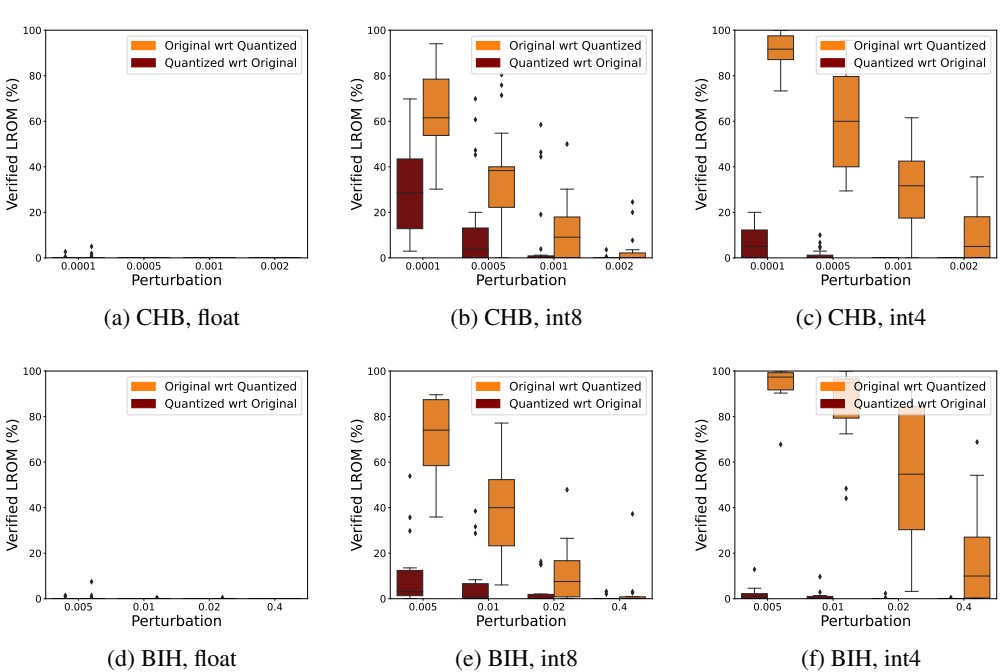

(a) CHB, float (b) CHB, int8 (c) CHB, int4

(d) BIH, float (e) BIH, int8 (f) BIH, int4

Figure 6: The box plots show verified LROM of the original and compact convolutional DNNs trained for all patients of the CHB-MIT and MIT-BIH datasets.

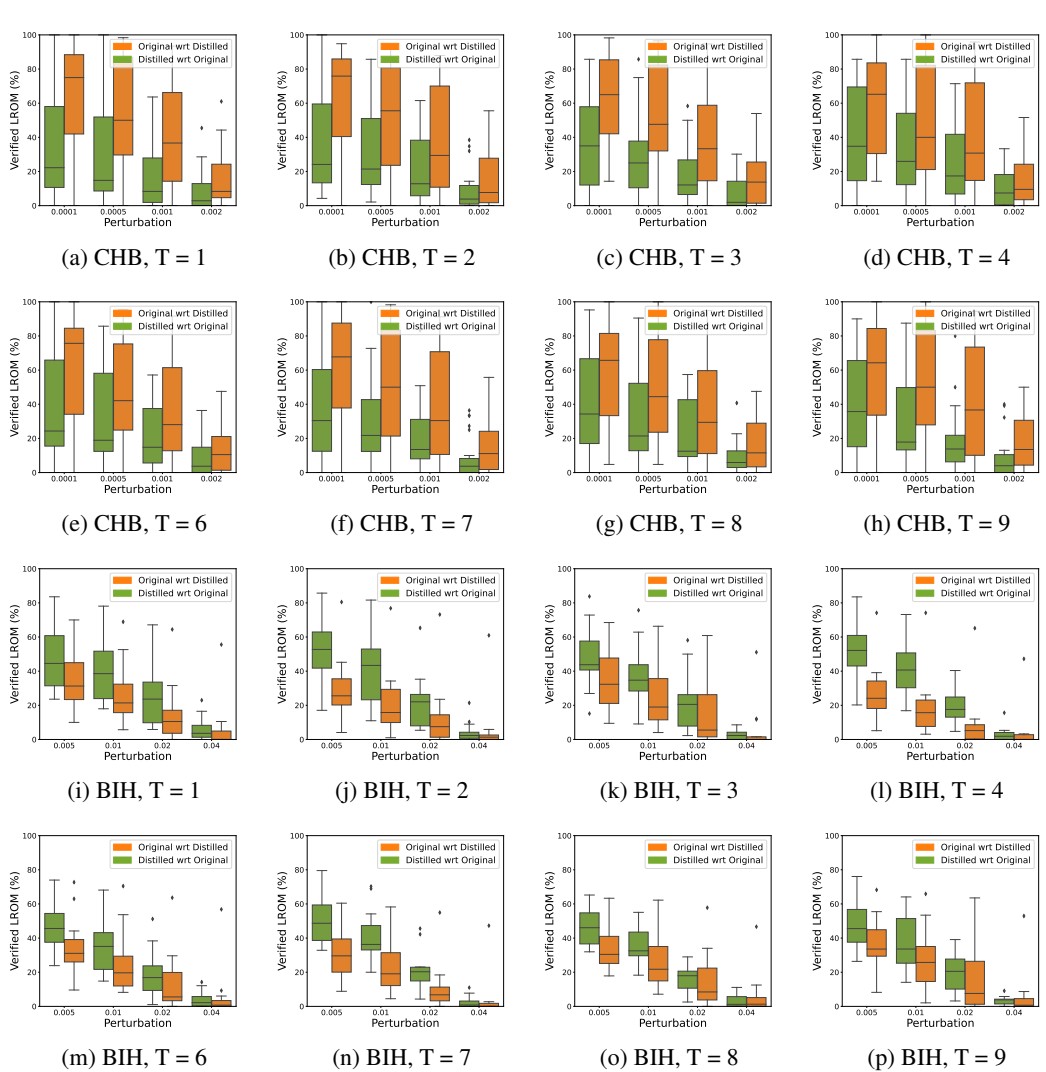

Figure 7: The box plots show verified LROM of the original and compact convolutional DNNs trained for all patients of the CHB-MIT and MIT-BIH datasets.

