# OpenReview forum: "Verified Relative Output Margins for Neural Network Twins"
_ICLR.cc/2025/Conference — Submitted to ICLR 2025_

### Official Review · Reviewer_eQgt · 2024-10-17

**Soundness:** 1
**Presentation:** 3
**Contribution:** 2
**Rating:** 3
**Confidence:** 4

**Summary:**

The paper presents a methodology for verifying agreement in neural network that try to approximate the same target function.
The proposed method focuses on the predicted likelihood-ratio between two classes (denoted as OM). In particular, it gives local bounds on the difference in log OM between the two networks (log LROM). The bounds are given for local neighborhoods, assuming that the correct label is known and constant in the region. The bounds on log LROM are obtained relaxing the exact optimization objective to an approximate one, solvable with linear programming.
The paper evaluates the proposed approach on a variety of scenarios, comparing distillation and quantization techniques, as well as evaluating the robustness of adversarial trained networks.

**Strengths:**

The paper is well structured, clear and straight to the point. The contribution is well laid out, and the examples for the empirical evaluation are rich, relevant and diverse.

**Weaknesses:**

The proposed metric of ROM represents the difference in prediction confidence, and does not necessarily relate to the agreement of the predictions. The LROM metric is only informative when the bounds do not contain zero. Moreover, even assuming a point has verifiable LROM, it might not be verifiable in the desired direction.

Moreover, the correctness of the LROM metric depends on the assumption that the correct label remains constant in the considered region. Therefore it is only sound for small enough neighborhoods. Looking at the empirical results, in many scenarios, the percentage of verifiable points rapidly drops to zero as the neighborhood size increases.
It is unclear if the proposed approach provides a qualitatively different result than a direct pointwise comparison of model predictions.

**Questions:**

The authors point out that having a strictly positive/negative LROM between two networks is a sufficient condition to ensure that one makes better predictions than the other. I understand that ROM as formulated has the advantage of being invariant to the choice of threshold, but I find it paints an incomplete picture, and can be misleading. In particular, two networks can have 100% compatible predictions, yet the ROM value could vary significantly. Similarly, the same ROM can represent drastically different situations. In fact, if the two networks have log OM of +0 and +10, we are comparing a coin toss to an extremely accurate predictor. This is not the same as having two extremely confident networks with log OM of +1000 and +1010, but both situations have log ROM of +10.

The discussion on the effectiveness of adversarial training is interesting, however I can't seem to find any plot/table of the results described in the main text.

I also would be interested in a more in depth discussion of the effects of the $\delta$ parameter on the metric. Clearly for small enough values, the approach reduces to a pointwise comparison of model predictions. To justify the approach, it should be shown (at least empirically) that evaluating LROM gives significantly different results.

The choice of the correct $\delta$ seems critical. If it is too large, the metric becomes meaningless and the number of verifiable points is likely to drop. Can you provide (at least) some heuristic to quantify a good value for $\delta$?
Also, have you considered making it a function of $x$, trying to estimate the largest local region where the correct label remains constant?

---

> ### Author Response · Authors · 2024-11-24
>
> We thank the reviewer for their review. We have addressed the comments below.
>
> >The proposed metric of ROM represents the difference in prediction confidence, and does not necessarily relate to the agreement of the predictions. The LROM metric is only informative when the bounds do not contain zero. Moreover, even assuming a point has verifiable LROM, it might not be verifiable in the desired direction.
>
> ROM represents relative difference in prediction confidence, but it is related to the agreement of the predictions. If LROM of N1 w.r.t. N2 is positive, it means that the OMs of N1 are consistently higher than the OMs of N2 and that is in *all directions* in the neighborhood we are considering.
>
> Even when LROM is negative, it still provides us with an informative quantitative measure. For instance, the compact models may not always be as good as their original counterparts. Therefore, the requirements from the application might be that the margins in the compact network may not be less than a certain percentage (80%) of the original network’s margins (i.e., negative LROM is accepted, but not less than a certain limit). Our framework provides guaranteed sound bounds in such cases.
>
> >Moreover, the correctness of the LROM metric depends on the assumption that the correct label remains constant in the considered region. … It is unclear if the proposed approach provides a qualitatively different result than a direct pointwise comparison of model predictions.
>
> We would like to also highlight that we do not make “the assumption that the correct label remains constant in the considered region”. Regardless of whether the correct label remains constant or not, our approach provides information about the LROM of two networks, indicating which network has a higher LROM than the other. Consider for example four cases for the margins of N1 and N2: (OM(N1),OM(N2))=(+,+), or (+,-), or (-,+), or (-,-). For (+,+), LROM will be positive if OM(N1)>OM(N2). For (+,-), LROM will be positive. For (-,+), LROM will be negative. For (-,-), positive LROM shows that OM(N1)>OM(N2).
>
> In addition, our experiments show that our approach is substantially more informative/accurate than looking individually (i.e., direct pointwise comparison of model predictions) into robustness properties of each network (“Comparison with Independent Analysis” and Figure 2 in Section 4.3).
>
> >... In fact, if the two networks have log OM of +0 and +10, we are comparing a coin toss to an extremely accurate predictor. This is not the same as having two extremely confident networks with log OM of +1000 and +1010, but both situations have log ROM of +10.
>
> Consider OM(N1)=0 and OM(N2)=10, which leads to LROM(N2,N1)=10. This is a *relative* condition between N1 and N2.
>
> Consider OM(N3)=1000 and OM(N4)=1010, which leads to LROM(N4,N3)=10. This is a *relative* condition between N3 and N4.
>
> Therefore, based on LROM(N2,N1) and LROM(N4,N3), we cannot draw any conclusions for (N1 or N2) vs (N3 or N4). However, if LROM(N3,N2)>0, then we know that LROM(N4,N1)>0, since we already know LROM(N2,N1)>0 and LROM(N4,N3)>0.
>
>
>
> >The discussion on the effectiveness of adversarial training is interesting, however I can't seem to find any plot/table of the results described in the main text.
>
> Due to the page limit, we only described the results and were unable to include a table/plot. Below is the summary of the results:
>
> | |Non-defended|PGD1|PGD3|
> |-|-|-|-|
> | Non-defended|-|0%|0%|
> | PGD1|57%|-|24%|
> | PGD3|56%| 38%|-|
>
> >... Clearly for small enough values, the approach reduces to a pointwise comparison of model predictions. To justify the approach, it should be shown (at least empirically) that evaluating LROM gives significantly different results.
>
> In our experiments (Section 4.3), we consider the adversarially-trained networks and non-defended networks and show that our framework captures that adversarially-trained models have larger LROM than the non-defended one. This is while both networks have similar “accuracy” and “certified accuracy” (i.e., the proportion of test samples for which a model is guaranteed to remain robust, i.e., correctly classify, within a specified perturbation radius). As such, our experiments show that LROM provides more insight into the difference between two networks, than “accuracy” and “certified accuracy”.
>
> >...Can you provide (at least) some heuristic to quantify a good value for $\delta$? …estimate the largest local region where the correct label remains constant?
>
> The value of $\delta$ is application-dependent, e.g., network architecture and dataset. One potential approach to estimate the largest local region (captured by $\delta$) is to iteratively perform our framework, similar to binary-search on $\delta$.
>
> We thank the reviewer again. We'd appreciate knowing if our clarifications/results have been satisfactory and if our clarified perspective might improve your evaluation of our work.

---

> > ### Comment · Reviewer_eQgt · 2024-11-26
> >
> > I believe my concerns might have been misunderstood, as the comment does not address them.
> > I will try to clarify what I meant and why the replies failed to change my opinion.
> >
> > > ROM represents relative difference in prediction confidence, but it is related to the agreement of the predictions.
> > [...]
> > Our framework provides guaranteed sound bounds in such cases.
> >
> > In my original comment I tried to point out that higher/lower OM does not relate to agreement in predictions. There are 4 scenarios: both models predict the correct class, both models predict the incorrect class, the models predict different classes.
> > The OM value, only serves to exclude one of the disagreement scenarios. Therefore, independently of OM, the models can disagree, agree and be correct, agree and be incorrect.
> >
> > > We would like to also highlight that we do not make “the assumption that the correct label remains constant in the considered region”.
> > [...]
> > For (-,+), LROM will be negative. For (-,-), positive LROM shows that OM(N1)>OM(N2).
> >
> > I understand that you do not need the assumption that the label remains constant in the domain to calculate the metrics. However, if the label is not constant in the domain the metrics you calculate are meaningless.
> >
> > >In addition, our experiments show that our approach is substantially more informative/accurate than looking individually (i.e., direct pointwise comparison of model predictions) into robustness properties of each network (“Comparison with Independent Analysis” and Figure 2 in Section 4.3).
> >
> > I believe the Figured referenced does not show what I asked. It shows how the joint analysis gives tighter bounds than the independent one. In the limit for $\delta=0$, the approach reduces to simply looking at the network predictions at a single point.
> > The claims of robustness should be evaluated by comparing the proposed approach to this much simpler and naive method.
> >
> > > Consider OM(N1)=0 and OM(N2)=10, which leads to [...] if LROM(N3,N2)>0, then we know that LROM(N4,N1)>0, since we already know LROM(N2,N1)>0 and LROM(N4,N3)>0.
> >
> > I understand that relative metrics are transitive. However my original comment was meant to highlight how this metric can be misleading. In your example, you consider the case when the scenarios with equal LROM and different logits are happening on different networks. My comment referred to evaluating LROM for different points but for the same networks. Consider a distilled network N1 and a baseline network N2. If LROM(N1)=1000 and LROM(N2)=1010, then i conclude that the distilled network is consistently underconfident compared to baseline. But does this matter? Does it make sense to treat this scenario the same as LROM(N1)=0 and LROM(N2)=10?
> >
> > >In our experiments (Section 4.3), we consider the adversarially-trained networks and non-defended networks and show that our framework captures that adversarially-trained models have larger LROM than the non-defended one.
> > [...]
> > The value of $\delta$ is application-dependent, e.g., network architecture and dataset. One potential approach to estimate the largest local region (captured by $\delta$) is to iteratively perform our framework, similar to binary-search on $\delta$.
> >
> > This results mentioned are only reported in the main text. The comparison to the limit for $\delta=0$ should be included in all results as a baseline. This ties to my other comments on the value of $\delta$. The proposed method has merits only if you are able to verify a large enough proportion of points for a large enough local radius, but not so large that the label cannot be assumed constant in the local region. This cannot be treated as a hyperparameter of secondary importance.
> >
> > I'm afraid that the authors' replies to some of the concerns I've raised only strengthen my confidence in the original score.
> > While I personally find the work and ideas interesting, I believe that the claims are not substantiated by a thorough evaluation. Moreover, the theoretical contributions offered are not sufficient to warrant acceptance by themselves.

---

> > > ### Author Response · Authors · 2024-11-30
> > >
> > > Thank you for the responses!
> > >
> > > >In my original comment I tried to point out that higher/lower OM does not relate to agreement in predictions. ... The OM value, only serves to exclude one of the disagreement scenarios. ...
> > >
> > > The state of the art techniques in the verification area  [1, 2] consider only correctly classified samples, to examine how networks behave in a perturbation region surrounding an actual point that is proven to be correctly classified. This is because if at least one of the networks makes an incorrect decision (negative margins), comparing the prediction of the networks is sufficient. Therefore, the only interesting scenario would be when both networks correctly classify the sample. The challenge is to compare the behaviors of the two networks in a neighborhood of a point where they both make a correct decision.
> > >
> > > We emphasize our claim in this paper (Lines 62-64): “LROM enables us to formally prove that a network consistently makes a correct decision every time the other network does, and it does so in the entire input region.”
> > >
> > > Let us reiterate the definition of LROM(N1,N2): min{OM(N1)-OM(N2)} on an entire neighborhood.
> > >
> > > To concretize the claim, suppose we could establish LROM(N1, N2) > 0 on a neighborhood around a sample “p”:
> > >
> > > * suppose a sample “s” in the neighborhood of “p” is classified incorrectly by both networks. We still have OM(N1) > OM(N2) for “s”, which means that N1 is closer to correctly classifying “s” than N2. So you can trust N1 if you did trust N2.
> > >
> > > * If a sample “s” in the neighborhood of “p” is classified correctly by N1, i.e., OM(N1)>0, and incorrectly by N2, i.e., OM(N2)<0, then again you can trust N1 if you did trust N2 as N1 is correct despite N2 making an incorrect prediction.
> > >
> > > * If a sample “s” in the neighborhood of “p” is classified incorrectly by N1 and correctly by N2, then N1 is doing worse than N2. Since N1 classifies the “s” incorrectly, then OM(N1)<0. Also, N2 classifies the sample “s” correctly implies OM(N2) > 0. As a result, the lower bound LROM(N1,N2) on the neighborhood has to be negative, which is excluded by our assumption LROM(N1,N2) > 0.
> > >
> > > * If sample “s” in the neighborhood is classified correctly by both networks with LROM(N1,N2)>0, then we know that the ratio of probabilities associated to the correct decision for “s” by N1 is larger than the ratio of probabilities associated by N2 for the same correct decision on “s”. Hence, you can trust N1.
> > >
> > > Again, showing LROM(N1,N2)>0 is useful because it means N1 (the network checked against a reference N2) is at least as correct as N2 (the reference network), even when margins of N1 and N2 are negative. We do not aim to show that N1 or N2 make the correct decisions on a neighborhood (i.e., have a constant label). That can be easily checked separately for N1 and N2 on the same neighborhoods using existing works. Our approach can show, on a whole neighborhood, that an implication holds. Namely, each time N2 (i.e., the reference network) makes the correct decision then N1 (the checked network) does also make the correct decision. Indeed, this excludes the case where the checked network N1 makes (in some points in the neighborhood) the wrong decision despite the reference network N2 making the right one. That is the whole point! If you trust the reference N2 enough (even in regions where it is not robust), then our framework can tell you N1 will make as good as or better decisions.
> > >
> > >
> > > [1] An abstract domain for certifying neural networks, Proceedings of the ACM on Programming Languages (PACMPL), 2019.
> > >
> > > [2] VNN: Verification-friendly neural networks with hard robustness guarantees. In the International Conference on Machine Learning (ICML), 2024.
> > >
> > > > ... However, if the label is not constant in the domain the metrics you calculate are meaningless.
> > >
> > > We do not agree the metric is meaningless when the labels are not constant. Let us reiterate the definition of LROM(N1,N2): min{OM(N1)-OM(N2)} on an entire neighborhood. Note that OM(N1) and OM(N2) exactly capture the actual predictions by network N1 and N2, respectively. If OM(N1)>0, the prediction is correct by N1. Similarly, If OM(N2)>0, the prediction is correct by N2. Therefore, if LROM(N1,N2)>0, this means that OM(N1)>OM(N2). Therefore, if you trust N2, then you can trust N1.
> > >
> > > >My comment referred to evaluating LROM for different points but for the same networks. …
> > >
> > > We would like to highlight that our work is to find **relative margins between two neural networks**, given a common input point, as the title of our paper suggests. The example from the reviewer discusses **relative margins between two input points**, which is a claim we have not made.
> > >
> > > Thank you for taking the time to review our work. We would appreciate your feedback on whether our clarifications regarding the contribution have addressed your concerns and whether they might positively influence your evaluation and confidence in our work.

---

> > > > ### Comment · Reviewer_eQgt · 2024-12-01
> > > >
> > > > At this point I am convinced the authors are deliberately avoiding addressing the questions (both mine and of other reviewers).
> > > >
> > > > I will summarize briefly.
> > > >
> > > > To address my comment that LROM does not relate to agreement in prediction, you say that it does when you are only considering cases where both networks make the correct decision. I agree, and that is exactly the concern. It only is useful when you trust a reference AND you can bound the ROM on the correct side.
> > > > I understand that it is useful, but you should recognize that its applicability is limited to specific scenarios.
> > > >
> > > > You do not address my comment that the label must be constant in the local region. You say that OM>0 means that the prediction is correct. This is true if the true label is constant. Even if you trust a reference, you can only trust a network with positive LROM **if** you are confident the true label is not changing in the region.
> > > >
> > > > I never claimed you are comparing margins between two points. I simply pointed out how the same result of the evaluation (same LROM between the networks) may represent drastically different scenarios. Moreover, you are comparing LROM at multiple points, and treating all equally, which provides misleading global aggregate metrics.
> > > >
> > > > The other points I have raised have been completely ignored.
> > > >
> > > > Given the (likely deliberate) avoidance in addressing the comments and the little time remaining, I don't see this discussion going any further.

---

> ### Author Response · Authors · 2024-12-01
>
> >At this point I am convinced the authors are deliberately avoiding addressing the questions (both mine and of other reviewers). I will summarize briefly.
>
> We focused on the main criticisms for brevity and for discussion coherence.
>
> >To address my comment that LROM does not relate to agreement in prediction, you say that it does when you are only considering cases where both networks make the correct decision. I agree, and that is exactly the concern. It only is useful when you trust a reference AND you can bound the ROM on the correct side. I understand that it is useful, but you should recognize that its applicability is limited to specific scenarios.
>
> This is inaccurate. Canonical definition of local robustness requires networks to maintain constant labels locally. Networks need not maintain constant predictions in general. If you show LROM(N1,N2)>0, then you know N1 will make the correct prediction each time N2 does. In our previous response, we analyze all four cases (for N1/N2 making correct/incorrect decisions) and show LROM is meaningful, despite the signs of OMs. We refer the reviewer to our previous answer.
>
> >You do not address my comment that the label must be constant in the local region. You say that OM>0 means that the prediction is correct. This is true if the true label is constant. Even if you trust a reference, you can only trust a network with positive LROM if you are confident the true label is not changing in the region.
>
> Indeed, there has been a misunderstanding about predicted labels and actual / ground-truth labels. **Local robustness** requires constant labels on considered balls (typically centered on correctly labeled samples). As the reviewer certainly agrees, we did not invent it. **LROM (Local Relative Output Margins) is also local**. However, in case two networks do not have constant predictions throughout a ball, we do not give up the comparison like current techniques. If you show LROM(N1,N2) > 0 (observe we can choose other thresholds for flexibility), then you know N1 will make the correct prediction (matching the should-be-constant-label) each time N2 does. If you use N2 as a reference (despite its occasional violations of local robustness) then you know N1 maintains the correct label at least each time N2 does.
>
> >I never claimed you are comparing margins between two points. I simply pointed out how the same result of the evaluation (same LROM between the networks) may represent drastically different scenarios. Moreover, you are comparing LROM at multiple points, and treating all equally, which provides misleading global aggregate metrics.
>
> That is what we understood from reviewer's previous comment: _“My comment referred to evaluating **LROM for different points but for the same networks**. …”_
>
> LROM(N1,N2) is each time computed for a single ball for both N1 and N2 (just like local robustness is checked for a network, a single ball at a time).
>
> We do not average the margin differences over the samples, rather we count the number of samples for which LROM(N1,N2)>0 vs LROM(N2,N1)>0. Therefore, the claim is not well founded.
>
> >The other points I have raised have been completely ignored.
>
> We now discuss the delta aspect, which we have not addressed in our previous response for discussion coherence and for space limitation.
>
> >I believe the Figured referenced does not show what I asked. It shows how the joint analysis gives tighter bounds than the independent one. In the limit for $\delta=0$, the approach reduces to simply looking at the network predictions at a single point.
> The claims of robustness should be evaluated by comparing the proposed approach to this much simpler and naive method.
>
> >The results mentioned are only reported in the main text. The comparison to the limit for $\delta=0$ should be included in all results as a baseline. This ties to my other comments on the value of $\delta$. The proposed method has merits only if you are able to verify a large enough proportion of points for a large enough local radius, but not so large that the label cannot be assumed constant in the local region. This cannot be treated as a hyperparameter of secondary importance.
>
> First, we only have the results in the text because other forms would not add much.
>
> We maintain that pointwise comparison is different from pointwise guarantees on neighborhoods: it only concerns specific samples. Borrowing from Dijkstra's statement that testing may only exhibit bugs, never show their absence, we highlight that working with concrete points can only hope to find counter-examples to "checked N1 makes the right decision each time reference N2 does". You cannot show their absence.
>
> >Given the (likely deliberate) avoidance in addressing the comments and the little time remaining, I don't see this discussion going any further.
>
> We have initially focused the discussion on LROM for brevity and for discussion coherence. We have now briefly addressed the remaining parts.

---

> > ### Comment · Reviewer_eQgt · 2024-12-02
> >
> > >We have now briefly addressed the remaining parts
> >
> > You have not.
> > I maintain my score

---

> > > ### Author Response · Authors · 2024-12-02
> > >
> > > We respectfully disagree, we answered within the limited space. The reviewer has also not responded to our clarifications.
> > >
> > > We are confident with our answers and are happy the discussion will be public for the community to judge.

---

### Official Review · Reviewer_c8iL · 2024-10-31

**Soundness:** 3
**Presentation:** 2
**Contribution:** 2
**Rating:** 3
**Confidence:** 4

**Summary:**

This manuscirpt define the notion of Relative Output Margin (ROM) and Local ROM (LROM) to compare network twins. Specifially, LROM > 0 means one network can consistently outperform another one in the vicinity of a given points. a theorem is provided the bound the LROM. Experiments on for datasets with 7-layer MLP are conducted to show the effectiveness of proposed LROM.

**Strengths:**

- A novel notion of Relative Output Margin is proposed.

- The organization (not writing) is very well.

- Experiments on multiple experiments are presented to show the interesting property of ROM.

**Weaknesses:**

## Major

- The conclution is unclear.
     - In my opinion, the main theorem (Theorem 3.1) solely establish that there indeed exists upper and lower bounds for any LROM. If it is, the significance of this theorem is limited and LROM cannot be linked to the generalizability of DNNs.
     - There is no clear conclusion for experiments. What do the experiments show?

- In the experiments, only a small 7-layer MLP is used, which hardly give a good predition for CIFAR-10, which makes a limited contribution. Can the authors provides the results at least on CNNs like VGG or ResNet?

- In the experiments, "We exclusively focus on **correctly** classified samples" (Line 299). For me, a big application for ROM is to measure the uncertainty of predictions. If we have already know that the predictions is correct, there is no point to use this technique.

## Minor

- The writing can be improved.
    - In Introduction, more words are needed to breifly introduce the theoretical and empirical work ( now only 6 lines from Line 61-66), which will make this paper more clear.
    - The notation are too complicated and can be simplified.

- It is better to number equations.

- use $\max$ and $\min$ to replace $max$ and $min$ in the equations.

- Line 215: "$-\mathcal{R}\geq \mathcal{M}$" -> "$-\mathcal{R} \geq -\mathcal{M}$"

**Questions:**

See Weaknesses.

---

> ### Author Response · Authors · 2024-11-23
>
> We acknowledge the reviewer’s efforts in assessing our work and are grateful for their feedback. We have addressed the comments and inquiries below.
>
> > The conclution is unclear. In my opinion, the main theorem (Theorem 3.1) solely establish that there indeed exists upper and lower bounds for any LROM. If it is, the significance of this theorem is limited and LROM cannot be linked to the generalizability of DNNs.
>
> Our approach addresses a very fundamental and interesting gap (Lines 62-64): “LROM enables us to formally prove that a network consistently makes a correct decision every time the other network does, and it does so in the entire input region.” This property is *on an entire input region*, hence the connection to generalizability.
>
> On the other hand, in our experiments (Section 4.3), we consider the adversarially-trained networks and non-defended networks and show that our framework captures that adversarially-trained models have larger LROM than the non-defended one. This is while both networks have similar “accuracy” and “certified accuracy” (i.e., the proportion of test samples for which a model is guaranteed to remain robust, i.e., correctly classify, within a specified perturbation radius). Therefore, LROM provides more insight into generalizability and robustness.
>
>
> >There is no clear conclusion for experiments. What do the experiments show?
>
> (1) Our experiments investigate the LROM between different models and provide examples of how our approach can be used to compare two networks (original vs distilled/quantized/pruned/VNN). This is shown in Section 4.3 and Figure 1 and Figure 3. However, they are not intended to draw conclusions about any specific model/network.
>
> (2) Our experiments show that LROM provides more insight into generalizability and robustness, than “accuracy” and “certified accuracy” (“Adversarially-Trained Models” in Section 4.3).
>
> (3) Our experiments show that our approach is substantially more informative/accurate than looking individually into robustness properties of each network (“Comparison with Independent Analysis” and Figure 2 in Section 4.3).
>
>
> >In the experiments, only a small 7-layer MLP is used, which hardly give a good predition for CIFAR-10, which makes a limited contribution. Can the authors provides the results at least on CNNs like VGG or ResNet?
>
> (1) We have based our experiments for the MNIST and CIFAR-10 datasets on neural networks sourced from the state-of-the-art studies in verification of neural networks [1, 2] in Sections 4.3.1 and 4.3.2.
>
> (2) The neural networks for the CHB-MIT and MIT-BIH datasets in Sections 4.3.3 and 4.3.4 are CNNs, adopted from a recent ICML paper [3]. In addition, we have already provided further experiments with CNNs on MNIST and CIFAR-10 datasets in the appendix (Appendix A.4.1, Figures 4 and 5 in the same appendix).
>
>
> [1] Proof transfer for fast certification of multiple approximate neural networks, Proceedings of the ACM on Programming Languages (PACMPL), 2022.
>
> [2] An abstract domain for certifying neural networks, Proceedings of the ACM on Programming Languages (PACMPL), 2019.
>
> [3] VNN: Verification-friendly neural networks with hard robustness guarantees, International Conference on Machine Learning (ICML), 2024.
>
>
> >In the experiments, "We exclusively focus on correctly classified samples" (Line 299). For me, a big application for ROM is to measure the uncertainty of predictions. If we have already know that the predictions is correct, there is no point to use this technique.
>
> Even when we know the predictions for the given sample are correct, the reasoning in the neighborhood of the sample is still relevant. This is the fundamental setting in “adversarial examples” [1]. An input sample may be correctly classified, but small perturbations can lead to misclassification of the input sample.
>
> It is common in the verification area to consider only correctly classified samples, to examine how networks behave in a perturbation region surrounding an actual point that is proven to be correctly classified [2]. However, this constraint can easily be removed from the analysis.
>
> [1] Adversarial examples in the physical world, Artificial intelligence safety and security, 2018.
>
> [2] An abstract domain for certifying neural networks, Proceedings of the ACM on Programming Languages (PACMPL), 2019.
>
> >Line 215: "$-\mathcal{R} \geq \mathcal{M}$" -> "$-\mathcal{R} \geq -\mathcal{M}$"
>
>
> After revisiting the derivation, we are confident that the original equation presented in our paper is correct (please see the proof of Theorem 3.1 in Appendix A.1). We welcome any further discussion or clarification to ensure mutual understanding and would be happy to provide additional details or insights if needed.
>
>
> We appreciate your thoughts and hope that our response has clarified our perspective. We also hope that this clarification may lead to an improved score from the reviewer.

---

> > ### Comment · Reviewer_c8iL · 2024-11-27
> >
> > Thank you for your reply. However, my concerns on theorem, empirical study, and limited datasets have not been fully addressed by potential insights of LROM on robustness. I would like to keep my rating.

---

> ### Author Response · Authors · 2024-11-29
>
> >Thank you for your reply. However, my concerns on theorem, empirical study, and limited datasets have not been fully addressed by potential insights of LROM on robustness. I would like to keep my rating.
>
> We thank the reviewer for their response. We would like to reiterate that, similar to the state of the art studies in the domain, we have used four datasets, MNIST, CIFAR-10, CHB-MIT, and MIT-BIH, out of which two are for real-world applications. We have considered numerous baselines for quantization, pruning, distillation, and VNNs, from top AI/ML conferences [1, 2]. Therefore, given the limited space, we believe the paper is sufficiently well supported,by a large number of experiments, including those presented in the supplementary material.
>
> [1] An abstract domain for certifying neural networks, Proceedings of the ACM on Programming Languages (PACMPL), 2019.
>
> [2] VNN: Verification-friendly neural networks with hard robustness guarantees. In the International Conference on Machine Learning (ICML), 2024.
>
> Moreover, we would like to point out certain claims made by the reviewer that are inaccurate:
>
> > In the experiments, "We exclusively focus on correctly classified samples" (Line 299). For me, a big application for ROM is to measure the uncertainty of predictions. If we have already know that the predictions is correct, there is no point to use this technique.
>
>
> As we discussed and previously highlighted, this is how the evaluation is conducted in the state-of-the-art works [1, 2].
>
> On the other hand, the reviewer seems to not be accustomed to the adversarial examples and robust certification domains and does not differentiate between pure prediction performance and robustness.
>
> As we previously mentioned, “An input sample may be correctly classified, but small perturbations can lead to misclassification of the input sample.”
>
> [1] An abstract domain for certifying neural networks, Proceedings of the ACM on Programming Languages (PACMPL), 2019.
>
> [2] VNN: Verification-friendly neural networks with hard robustness guarantees. In the International Conference on Machine Learning (ICML), 2024.
>
> > Line 215: "$-\mathcal{R} \geq \mathcal{M}$" -> "$-\mathcal{R} \geq -\mathcal{M}$"
>
> Please note the difference between $\mathcal{R^{N1\mid N2}}$ and $\mathcal{R^{N2\mid N1}}$. The theorem explains that  $\mathcal{R^{N1\mid N2}} \leq \mathcal{M^{N1\mid N2}}$ and $-\mathcal{R^{N2\mid N1}} \geq \mathcal{M^{N1\mid N2}}$.
>
> The reviewer has initially claimed that the theorem is not correct (which cannot be taken lightly) and we appreciate it if the reviewer could help us identify the issue; or adjust their score otherwise.
>
> We thank the reviewer for their time. We'd appreciate knowing if our clarifications on contribution have satisfied you and if our clarified perspective might improve your evaluation of our work and your confidence in our work.

---

### Official Review · Reviewer_dty1 · 2024-11-03

**Soundness:** 3
**Presentation:** 3
**Contribution:** 3
**Rating:** 6
**Confidence:** 4

**Summary:**

This paper presents a framework for comparing two neural network classifiers with shared input and output domains. The goal is to analyze these "neural network twins" through the concept of Relative Output Margin (ROM), a metric indicating the confidence with which one network outperforms the other within a defined input region. Specifically, the framework formalizes and verifies "Local Relative Output Margins" (LROMs), allowing for the computation of provable bounds that indicate which network consistently makes correct decisions across input variations. This is crucial in applications where compact, optimized versions of networks are used, such as in medical device deployment, where safety-critical tasks like seizure and arrhythmia detection require guaranteed performance reliability.

The experiments in the paper evaluate the proposed Relative Output Margin (ROM) and Local Relative Output Margin (LROM) framework by testing it on four datasets: MNIST, CIFAR-10, CHB-MIT (EEG data for epilepsy detection), and MIT-BIH (ECG data for arrhythmia detection). The experiments focus on verifying LROM across different pairs of neural networks: original, pruned, quantized, and distilled versions. Across datasets, the experiments demonstrate that the LROM framework effectively captures the comparative performance and robustness of different network types under a defined perturbation range.

**Strengths:**

- The paper’s primary contribution lies in defining and formalizing ROM and LROM. This enables a provable, quantitative comparison between neural networks for applications requiring high reliability and safety.

- The study evaluates the proposed framework across multiple datasets, including standard and specialized medical data, supporting the generalizability and robustness of LROM as a comparative measure.

**Weaknesses:**

- The LROM optimization framework requires handling complex linear programming tasks, which may limit scalability for larger networks. It would be better to test the framework further on larger neural networks, such language models.

- It may be challenging to interpret the evaluated measures due to the technical intricacies involved in LROM computation.

**Questions:**

- It would be better to conduct a comparison with existing adversarial robustness metrics and methods. How do the proposed measures differ from the existing adversarial robustness measures?

- What would be the applications of the framework beyond network twins (similar architectures with compact versions), such as varied neural architectures?

---

> ### Author Response · Authors · 2024-11-24
>
> We acknowledge the reviewer’s efforts in assessing our work and are grateful for their feedback. We have addressed the comments and inquiries below.
>
>
> >The LROM optimization framework requires handling complex linear programming tasks, which may limits calability for larger networks. It would be better to test the framework further on larger neural networks, suchl anguage models.
>
> We would like to highlight that this limitation is already explicitly mentioned in our paper (Lines 533–535). Scalability is indeed a common challenge in the entire formal verification area (not particular to this work)  [1], but it is the price to pay for providing formal guarantees, which is necessary in safety-critical applications, e.g., in the medical domain.
>
> That being said, we have based our experiments for the MNIST and CIFAR-10 datasets on neural networks sourced from the state-of-the-art studies in verification of neural networks [1, 2] in Sections 4.3.1 and 4.3.2.
> The neural networks for the CHB-MIT and MIT-BIH datasets in Sections 4.3.3 and 4.3.4 are CNNs, adopted from a recent ICML paper [3].
>
>
> [1] Proof transfer for fast certification of multiple approximate neural networks, Proceedings of the ACM on Programming Languages (PACMPL), 2022.
>
> [2] An abstract domain for certifying neural networks, Proceedings of the ACM on Programming Languages (PACMPL), 2019.
>
> [3] VNN: Verification-friendly neural networks with hard robustness guarantees. In the International Conference on Machine Learning (ICML), 2024.
>
>
>
> >It may be challenging to interpret the evaluated measures due to the technical intricacies involved in LROM computation.
>
> While we understand the reviewer’s point of view, this is the nature of formal methods and verification domains, where the properties need to be defined and proved formally to be able to provide hard guarantees. In our final version, should the space permit, we will provide more intuitions for the formal content.
>
>
> >It would be better to conduct a comparison with existing adversarial robustness metrics and methods. How do the proposed measures differ from the existing adversarial robustness measures?
>
> Adversarial robustness metrics are designed to quantify a model's ability to withstand adversarial attacks, but they do not guarantee the results. On the other hand, formal verification provides sound results, meaning it guarantees the extent to which a network is robust against perturbations. Our work is based on formal verification, where we provide formal guarantees on the relationship between the output margins of two networks.
>
> In our experiments (Section 4.3), we consider the adversarially-trained networks and non-defended networks and show that our framework captures that adversarially-trained models have larger LROM than the non-defended one. This is while both networks have similar “accuracy” and “certified accuracy” (i.e., the proportion of test samples for which a model is guaranteed to remain robust, i.e., correctly classify, within a specified perturbation radius). As such, our experiments show that LROM provides more insight into the difference between two networks, than “accuracy” and “certified accuracy”.
>
> On the other hand, currently, there is no adversarial technique to perform such analyses jointly (for the same perturbation) for two networks. At the same time, performing the analysis independently would lead to excessive over-approximation and lead to inconclusive results (as shown in our experiments).
>
> Our experiments show that our approach is substantially more informative/accurate than looking individually into robustness properties of each network (“Comparison with Independent Analysis” and Figure 2 in Section 4.3).
>
>
> >What would be the applications of the framework beyond network twins (similar architectures with compact versions), such as varied neural architectures?
>
> Our framework and approach is by no means limited to “similar architectures with compact versions”. The only requirement is to have the same input/output domains and that is necessary (because we need to check the outputs of the two networks for the same input). Please see Lines 042-045: “In this work, we focus on neural network twins, i.e., two neural networks trained for the same learning/classification task, with the same input and output domains, but not the same weights and/or architectures.”
>
>
> We thank the reviewer for their time. We'd appreciate knowing if our clarifications on contribution have satisfied you and if our clarified perspective might improve your evaluation of our work and your confidence in our work.

---

> > ### Comment · Reviewer_dty1 · 2024-11-24
> >
> > Thanks for the authors' responses! They have addressed most of my concerns. Thus, I would like to increase my score accordingly.

---

> ### Author Response · Authors · 2024-11-29
>
> We thank the reviewer for their response and their decision to increase the score of our paper.

---

### Official Review · Reviewer_tfBm · 2024-11-04

**Soundness:** 3
**Presentation:** 3
**Contribution:** 1
**Rating:** 3
**Confidence:** 3

**Summary:**

This paper is interested in certifying the output of a given network w.r.t another one, targeting use-cases like pruning/distillation/quantization, in the goal of demonstrating that the pruned/distilled/quantized network exhibits not only similar accuracy over the train set, but even consistent decisions around the same local regions around each training example.

To do so, they introduce a novel measure called Relative Output Margin (ROM), with is the ratio between the Output Margin (OM) of two networks at a given input point. Finally, by taking the minimum of the ROM over a whole infinity-norm ball centered around a given point, they define the Local ROM (LROM).

Computing the exact LROM is a hard problem, but its linear relaxation for feed-forward ReLU networks is tractable and yields a lower bound, which is sufficient for the purpose of certification.

The algorithm is tested in the context of pruning, quantization, and distilled networks, on two images datasets (MNIST, CIFAR-10), and two tabular/signal datasets (CHB-MIT and MIT-BIH).

**Strengths:**

### Originality

It think the main originality of the paper lies in considering the joint optimization of the margins of two networks. Intuitively, one can understand why this is a superior approach compared to optimizing separately the bounds and trying to aggregate them (although the optimization problem is now double the dimension), as shown in Figure 2.

Focusing on the context of pruned/distilled/quantized networks is also relevant, re-targeting the (sometimes too ambitious) conventional goal of certifying a given network into just showing that some networks is not much worse than an other.

### Clarity and soundness

The paper is clear overall, proofs look correct.

**Weaknesses:**

### Novelty

My main source of concern is the novelty. Computing certificates for ReLU-based networks is a well established methodology that relies on various linear relaxation (as used in this paper), or interval propagation.

This paper utilizes these tools, and the only novelty is to consider a joint optimization of the difference of logits for two networks, instead of a single one, which I consider a straightforward extension departing from existing methods.

Therefore the main contribution of the paper is introducing the OM, ROM and LROM measures, and using the whole Section 2 to deal with rather trivial considerations.

All the proofs of appendix A.1 are a trivial consequence of manipulating the log of probability ratios, to end-up with a simple difference of logits. This is colloquially called the *margin*, not to be confused with the Output Margin (OM) measure that authors introduce, without clear motivation.

In most papers for NN certification, this margin is analyzed, reported  or even optimized (using Hinge loss) in a straightforward manner, without bothering highlighting the link with output probabilities. In this regard, the theoretical contribution appears rather shallow IMHO.

Dropping this narrative would even allow to use the method of the paper outside the context of classification. Even the concept of “twin networks” looks overkill to simply describe networks operating over the same input/output spaces.

**Questions:**

1) LROM is assymmetric, i.e inverting DNN N1 and N2 yields a different bound. Can you comment on this property? Is this something desirable? What if LROM(N1, N2) equals some value, and LROM(N2, N1) equals another, is there something to interpret here (qualitatively)?

2) When LROM is very negative (i.e logits difference is huge) which implies that probabilities are near-zero, not too far away from machine precision zero, do you expect this value to carry meaningful information?

3) Can you clarify use-cases in which LROM is useful for practioner?

---

> ### Author Response · Authors · 2024-11-23
>
> We thank the reviewer for their time and effort in reviewing our work, and we appreciate their valuable feedback.
>
> Weaknesses:
>
> >This paper utilizes these tools, and the only novelty is to consider a joint optimization of the difference of logits for two networks, instead of a single one, which I consider a straightforward extension departing from existing methods.
> Therefore the main contribution of the paper is introducing the OM, ROM and LROM measures, and using the whole Section 2 to deal with rather trivial considerations.
>
> Our approach answers a very fundamental and interesting question: “Given two compatible networks and an input region, is it possible to formally prove that one network consistently makes a correct decision every time the other network does, in the entire input region.” This has not been done before.
>
> >All the proofs of appendix A.1 are a trivial consequence of manipulating the log of probability ratios, to end-up with a simple difference of logits. This is colloquially called the margin, not to be confused with the Output Margin (OM) measure that authors introduce, without clear motivation.
>
> We would like to highlight that our main novelty and contribution is the *sound* analysis of the LROM for two neural networks. The manipulation referred to by the reviewer allows us to perform this analysis without any approximation (in the objective function) and in a formally sound fashion, which can be formulated as linear programming.
>
> We use *output* margins, i.e. margin in the output domain, not to be confused with input margins or margin (e.g., as in SVMs).
>
> >In most papers for NN certification, this margin is analyzed, reported or even optimized (using Hinge loss) in a straightforward manner, without bothering highlighting the link with output probabilities. In this regard, the theoretical contribution appears rather shallow IMHO.
>
> Our paper aims at analysis of relative margins. Optimization of margins is outside the scope of this work.
>
> Questions:
>
> >LROM is assymmetric, i.e inverting DNN N1 and N2 yields a different bound. Can you comment on this property? Is this something desirable? What if LROM(N1, N2) equals some value, and LROM(N2, N1) equals another, is there something to interpret here (qualitatively)?
>
> We answer this question in Theorem 3.1, also mentioned in Lines 214-215 of the main paper. Essentially, because of the relaxation/approximations in formal methods, the *exact* value of LROM(N1, N2) is bounded from below by LROM(N1,N2), which is found by our approach, and bounded from above by -LROM(N2, N1). Therefore, not only can we provide a safe lower-bound on the exact LROM(N1, N2), but also a safe upper-bound.
>
> >When LROM is very negative (i.e. logits difference is huge) which implies that probabilities are near-zero, not too far away from machine precision zero, do you expect this value to carry meaningful information?
>
> We would appreciate it if the reviewer could elaborate on the question. However, we should highlight that very negative LROM does not necessarily imply that probabilities are near zero.
>
> >Can you clarify use-cases in which LROM is useful for practioner?
>
> Let us highlight the motivation and use-case we discuss for practitioners in Lines 29-36 (first paragraph of Introduction): “In the medical domain, for instance, neural networks can enable implantable and wearable devices to detect cardiac arrhythmia (Sopic et al., 2018a) or epileptic seizures (Baghersalimi et al., 2024) in real time. However, due to their limited computing resources, such devices often adopt the compact networks corresponding to the original medical-grade networks. It is vital for the compact network to reliably detect cardiac abnormalities/seizures, as lack of reliable decisions can jeopardize patients’ lives. Therefore, reasoning about the decisions made by the compact network w.r.t. to an original/reference network is vital for the safe deployment of the compact networks.”
>
> Our approach answers this very fundamental question: “Given these two compatible networks and an input region, is it possible to formally prove that one network consistently makes a correct decision every time the other network does, in the entire input region.”
>
> We thank the reviewer for their time. We'd appreciate knowing if our clarifications on contribution have satisfied you and if our clarified perspective might improve your evaluation of our work and your confidence in our work.

---

> > ### Comment · Reviewer_tfBm · 2024-11-27
> >
> > Thank you for your clarifications.
> >
> > > “Given two compatible networks and an input region, is it possible to formally prove that one network consistently makes a correct decision every time the other network does, in the entire input region.”
> >
> > The entire input regions is still characterized by a set of balls centered around the train set. Do you believe this brings better guarantees than just monitoring the certified robustness (using labels) over the train set?
> >
> > > The manipulation referred to by the reviewer allows us to perform this analysis without any approximation (in the objective function) and in a formally sound fashion
> >
> > Sorry, but I believe this manipulation is rather straightforward.
> >
> > > We use output margins, i.e. margin in the output domain, not to be confused with input margins or margin (e.g., as in SVMs). [...] Optimization of margins is outside the scope of this work.
> >
> > I agree. I repeat what I said: "*All the proofs of appendix A.1 are a trivial consequence of manipulating the log of probability ratios, to end-up with a simple difference of logits. This is colloquially called the margin*". This is the standard practice in robustness certification, or in training with the Hinge loss.
> >
> > I don't understand why the LROM is important in the first place, this measure looks very *ad-hoc*. No formal justification is given on *why* that should be the correct measure to compare "compatible networks". You said "*is it possible to formally prove that one network consistently makes a correct decision every time the other network does, in the entire input region*"; I agree this is an interesting question, but there is no discussion on why LROM is the correct measure to answer this question.
> >
> > If the goal behind introducing LROM was just to obtain a difference of logits in the optimization problem of line 163, you just re-discovered the rational behind multi-class hinge loss. As I said, most papers in robustness certification use the margin, no need to introduce LROM as a contribution. You could simply start the paper by stating the optimization problem of line 163; that would be fine, skipping lines 99 to 160 entirely, that would look even more logical to me than introducing LROM.
> >
> > > We would appreciate it if the reviewer could elaborate on the question.  we should highlight that very negative LROM does not necessarily imply that probabilities are near zero
> >
> > LROM is essentially the logarithm of a ratio of probabilities. If the log is negative, then the ratio is near zero, which means that the numerator is negligible compared to the denominator. I am questioning the rational behind LROM as a measure (as discussed before) on this corner case.
> >
> > > Let us highlight the motivation and use-case we discuss for practitioners in Lines 29-36 (first paragraph of Introduction): [...]
> >
> > I am not questioning the motivation of your work, as I recognize its importance in both in the "*summary*" and the "*strengths*" section of my review. However I am questioning your solution. See below:
> >
> > > Our approach answers this very fundamental question: “Given these two compatible networks and an input region, is it possible to formally prove that one network consistently makes a correct decision every time the other network does, in the entire input region.”
> >
> > Sorry, but I am simply not convinced that this ad-hoc measure, that you obtain with a straightforward extension of existing LP relaxations, actually answer the question in a satisfactory manner.
> >
> > I do not see much a progress in the discussion, therefore I'd like to keep my score.

---

> > > ### Author Response · Authors · 2024-11-30
> > >
> > > Thank you for the prompt response!
> > >
> > > >The entire input regions is still characterized by a set of balls centered around the train set. Do you believe this brings better guarantees than just monitoring the certified robustness (using labels) over the train set?
> > >
> > > Indeed. The balls are centered around a test set, not the train set. This is well established when verifying robustness with formal guarantees on entire “balls”. This is not possible with testing or adversarial examples/training.
> > >
> > > At the same time, in our experiments (Section 4.3), we consider two adversarially-trained networks and a non-defended network and show that our framework captures that adversarially-trained models have larger LROM than non-defended ones. This is while both networks have similar “accuracy” and “certified accuracy".
> > >
> > > >Sorry, but I believe this manipulation is rather straightforward.
> > >
> > > The simplicity of the manipulation does not make it useless, nor does it warrant rejection. Formal verification of local robustness, a well-established area, often only looks at one single network (see for example [1] with 1000+ citations, or more recent work at ICML 2024 [2]). These studies are even simpler than our manipulation (because we consider and compare two networks), yet, it is commonly accepted because it shows robustness on balls centered on a test set. Here, the comparison is simple, but not as straightforward: we compare the ratios of probabilities of the correct/incorrect decisions in two networks. This allows us to determine whether a network is at least as robust as another. We are not aware of any work that considers this aspect.
> > >
> > > [1] AI2: Safety and Robustness Certification of Neural Networks with Abstract Interpretation, S&P, 2018.
> > >
> > > [2] VNN: Verification-friendly neural networks with hard robustness guarantees. ICML, 2024.
> > >
> > >
> > > >I don't understand why the LROM is important in the first place, this measure looks very ad-hoc.
> > >
> > > LROM is by no means ad-hoc. Let us reiterate the definition of LROM(N1,N2): min{OM(N1)-OM(N2)} on an entire neighborhood. In short, if LROM(N1,N2)>0, this means that OM(N1)>OM(N2). Therefore, if you trust N2, you can trust N1. Note that OM(N1) and OM(N2) exactly capture the actual predictions by network N1 and N2, respectively.
> > >
> > > To make our claim concrete, suppose we could establish LROM(N1, N2) > 0 on a neighborhood around a sample “p”:
> > >
> > > * suppose a sample “s” in the neighborhood of “p” is classified incorrectly by both networks. We still have OM(N1) > OM(N2) for “s”, which means that N1 is closer to correctly classifying “s” than N2. So you can trust N1 if you did trust N2.
> > >
> > > * If a sample “s” in the neighborhood of “p” is classified correctly by N1, i.e., OM(N1)>0, and incorrectly by N2, i.e., OM(N2)<0, then again you can trust N1 if you did trust N2 as N1 is correct despite N2 making an incorrect prediction.
> > >
> > > * If a sample “s” in the neighborhood of “p” is classified incorrectly by N1 and correctly by N2, then N1 is doing worse than N2. Since N1 classifies the “s” incorrectly, then OM(N1)<0. Also, N2 classifies the sample “s” correctly implies OM(N2) > 0. As a result, the lower bound LROM(N1,N2) on the neighborhood has to be negative, which is excluded by our assumption LROM(N1,N2) > 0.
> > >
> > > * If sample “s” in the neighborhood is classified correctly by both networks with LROM(N1,N2)>0, then we know that the ratio of probabilities associated to the correct decision for “s” by N1 is larger than the ratio of probabilities associated by N2 for the same correct decision on “s”. Hence, you can trust N1.
> > >
> > > We do not aim to show that N1 or N2 make the correct decisions on a neighborhood (i.e., have a constant label). That can be easily checked separately for N1 and N2 on the same neighborhoods using existing works. Our approach can show, on a whole neighborhood, that each time N2 (i.e., the reference network) makes the correct decision then N1 (the checked network) does also make the correct decision.
> > >
> > > >Sorry, but I am simply not convinced that this ad-hoc measure, that you obtain with a straightforward extension of existing LP relaxations, actually answer the question in a satisfactory manner.
> > >
> > > LROM is by no means ad-hoc, despite its simplicity. Let us reiterate the definition of LROM(N1,N2): min{OM(N1)-OM(N2)} on an entire neighborhood. Note that OM(N1) and OM(N2) exactly capture the actual predictions by network N1 and N2, respectively. If OM(N1)>0, the prediction is correct by N1. Similarly, If OM(N2)>0, the prediction is correct by N2. Therefore, if LROM(N1,N2)>0, this means that OM(N1)>OM(N2), on an entire neighborhood. Therefore, if you trust N2 (i.e., OM(N2)>0) and we know LROM(N1,N2)>0, then you can trust N1 (i.e., OM(N1)>OM(N2)>0). We are happy to include this in our paper.
> > >
> > > We appreciate the reviewer’s time and would appreciate knowing if our clarifications on the contribution have addressed your concerns and influenced your evaluation of our work.

---

> ### Comment · Reviewer_tfBm · 2024-12-02
>
> Thank you for your additional clarifications
>
> > The simplicity of the manipulation does not make it useless, nor does it warrant rejection.
>
> Indeed. But it does not warrant acceptance either. You are wasting a full page on it without clear advantage for the reader.
>
> >  These studies are even simpler than our manipulation (because we consider and compare two networks), yet, it is commonly accepted
>
> I don't believe acceptance and rejection in science should follow the same logic as "case-law" and "legal precedence". Especially when we factor in the novelty. Paper [1] is from 2018 and the novelty for that time makes it a stronger case than a submission in 2024.
>
> > We do not aim to show that N1 or N2 make the correct decisions on a neighborhood (i.e., have a constant label). **That can be easily checked separately for N1 and N2 on the same neighborhoods using existing works**. Our approach can show, on a whole neighborhood, that each time N2 (i.e., the reference network) makes the correct decision then N1 (the checked network) does also make the correct decision.
>
> (bold emphasis by me).
>
> I think you are really onto something with this last remark. It seems that this is a concern shared by other reviewers. Overall, it seems that we are likely unconvinced by LROM alone. If you argue that it "*can be easily checked separately for N1 and N2 on the same neighborhoods using existing works*" then I believe this is your responsibility to use these "existing works" and combine the results with LROM to *really check* that this additional information brings more insights than the raw certifiable robustness of N2 in the neighborhood of each train/test example.
>
> If you perform these experiments in a further re-submission it could make a compelling case for your work.

---

> > ### Author Response · Authors · 2024-12-02
> >
> > Thank you for your response!
> >
> > We would like to highlight that the reviewer has not responded to part of our rebuttal and their previous claim that LROM is a “very ad-hoc” measure. We consider our previous answer has been satisfactory.
> >
> > > You are wasting a full page on it without clear advantage for the reader.
> >
> > Yet the point is still misunderstood as witnessed by the comments about the ad-hoc nature of LROM. The “trivial consequence of manipulating the log of probability ratios [quoted from the reviewer]”, are just included for completeness, in the supplementary material, which has no limit.
> >
> > >I think you are really onto something with this last remark. It seems that this is a concern shared by other reviewers. Overall, it seems that we are likely unconvinced by LROM alone. If you argue that it "can be easily checked separately for N1 and N2 on the same neighborhoods using existing works" then I believe this is your responsibility to use these "existing works" and combine the results with LROM to really check that this additional information brings more insights than the raw certifiable robustness of N2 in the neighborhood of each train/test example.
> >
> > >If you perform these experiments in a further re-submission it could make a compelling case for your work.
> >
> > What can be easily checked with existing work is to independently ask whether a network is locally robust. Current approaches are too coarse to check the implication (N1 makes a correct prediction each time N2 has does a correct prediction), let alone to give tight bounds on the quotient of ratios with which predictions are made.
> >
> > We have already done experiments to compare performing such analysis independently and have shown that our approach is substantially more accurate/tight than looking individually into robustness properties of each network (“Comparison with Independent Analysis” and Figure 2 in Section 4.3).
> >
> > In addition, in our experiments (Section 4.3, “Adversarially-Trained Models”), we consider the adversarially-trained networks with PGD and non-defended networks and show that our framework captures that adversarially-trained models have larger LROM than the non-defended one. This is while both networks have similar “accuracy” and “certified accuracy”. As such, our experiments show that LROM provides more insights into the difference between two networks, than “accuracy” and “certified accuracy”.
> >
> > Despite the explanations in the paper and in our answers, we can only notice misunderstandings about what LROM adds compared to independently checking local robustness. We can compare networks on balls even when they are not locally robust there. We can even use thresholds to bound the ratios of their decisions wrt. to a robust prediction. No previous work does that.
> >
> > We thank the reviewer for the comments. LROM gives insights that are not possible to get with existing work. We shall think of ways to better clarify this.

---

### Official Review · Reviewer_mznd · 2024-11-12

**Soundness:** 2
**Presentation:** 2
**Contribution:** 2
**Rating:** 5
**Confidence:** 2

**Summary:**

I am pretty new to this field so I will try to summarize what I understand to be the key contributions of this paper. If I am wrong, please do let me know in the comments:

1. The authors introduce a new way to compare two neural networks (e.g. an original and a compressed version of the same net) by looking at their "relative output margins" = essentially comparing how confidently they make the same decisions.

2. They provide a formal verification framework that can prove, within a small neighborhood of a given input (like a small perturbation of an image), that one network will always make decisions at least as confidently as another network when they're both correct.

They demonstrate this is practically useful when:

a. Comparing original networks with their pruned/quantized/distilled versions
b. Analyzing medical AI systems where reliability is crucial
c. Understanding the relationship between regular and adversarially-trained models

The key innovation is that instead of trying to verify properties across all possible inputs (which would be intractable), they focus on small, local neighborhoods around specific inputs and use linear programming techniques to efficiently compute provable bounds on the networks' relative behavior in these regions.

**Strengths:**

The problem the authors are addressing is import and relevant:
How to formally compare two neural networks' decision confidence across many examples and not only at the given evaluation datapoints. This is especially relevant when we modify networks via e.g. quantization and pruning but need guarantees (in high stakes settings such as medicine).

The linear programming formulation makes the solution practical, which is good -- imho a method that can be practically applied is a key to wide adoption and impact.

I also appreciate the comparison to adversarially trained models.

**Weaknesses:**

I have a few concerns:

1. Scalability

1.1. linear programming can be expensive -- how well does this scale to larger networks?
1.2. small networks and small regions are shown in the paper. How well would this do on e.g. a ~100M parameter ViT and with larger regions?

2. How tight are the bounds? Do you have any experiments to demonstrate that? I would also be great to discuss worst-case scenarios with examples and develop some kind of a rudimentary case study of that.

3. Small regions
The small perturbation sizes used (0.001, 0.01) may not reflect real-world distortions. If I add a bit of a Gaussian noise to the whole image, I can easily get much higher delta.

4. Comparison to other linear programming based methods
I think there are other methods that use local approximations to calculate the difference between networks (I might be wrong on this), yet you don't compare to them?

5. Medical Application Claims
It would be good to compare your estimates to some sources of ground truth. Is that feasible?

**Questions:**

Included in weaknesses.

---

> ### Author Response · Authors · 2024-11-24
>
> We acknowledge the reviewer’s efforts in assessing our work and are grateful for their feedback. We have addressed the comments and inquiries below.
>
>
> >1. Scalability
> 1.1. linear programming can be expensive -- how well does this scale to larger networks? 1.2. small networks and small regions are shown in the paper. How well would this do on e.g. a ~100M parameter ViT and with larger regions?
>
> We would like to highlight that this limitation is already explicitly mentioned in our paper (Lines 533–535). Scalability is indeed a common challenge in the entire formal verification area (not particular to this work)  [1], but it is the price to pay for providing formal guarantees, which is necessary in safety-critical applications, e.g., in the medical domain.
>
>
> That being said, we have based our experiments for the MNIST and CIFAR-10 datasets on neural networks sourced from the state-of-the-art studies in verification of neural networks [1, 2] in Sections 4.3.1 and 4.3.2.
> The neural networks for the CHB-MIT and MIT-BIH datasets in Sections 4.3.3 and 4.3.4 are CNNs, adopted from a recent ICML paper [3].
>
>
> [1] Proof transfer for fast certification of multiple approximate neural networks, Proceedings of the ACM on Programming Languages (PACMPL), 2022.
>
> [2] An abstract domain for certifying neural networks, Proceedings of the ACM on Programming Languages (PACMPL), 2019.
>
> [3] VNN: Verification-friendly neural networks with hard robustness guarantees. In the International Conference on Machine Learning (ICML), 2024.
>
>
> >2. How tight are the bounds? Do you have any experiments to demonstrate that? I would also be great to discuss worst-case scenarios with examples and develop some kind of a rudimentary case study of that.
>
> While the tightness of the bounds is interesting, we would like to highlight that our approach is *sound*. This means that, when our framework concludes that  a network consistently makes a correct decision every time the other network does, in the entire input region, the conclusion is guaranteed to be correct.
>
> In terms of tightness, our experiments show that our approach is substantially more accurate/tight than looking individually into robustness properties of each network (“Comparison with Independent Analysis” and Figure 2 in Section 4.3).
>
>
> >3. Small regions The small perturbation sizes used (0.001, 0.01) may not reflect real-world distortions. If I add a bit of a Gaussian noise to the whole image, I can easily get much higher delta.
>
> Our experiments show that despite small regions, our framework can provide more insight into the difference between two networks, even with such small regions. In our experiments (Section 4.3), we consider the adversarially-trained networks and non-defended networks and show that our framework captures that adversarially-trained models have larger LROM than the non-defended one. This is while both networks have similar “accuracy” and “certified accuracy” (i.e., the proportion of test samples for which a model is guaranteed to remain robust, i.e., correctly classify, within a specified perturbation radius). As such, our experiments show that LROM provides more insight into the difference between two networks, than “accuracy” and “certified accuracy”.
>
> >4. Comparison to other linear programming based methods I think there are other methods that use local approximations to calculate the difference between networks (I might be wrong on this), yet you don't compare tothem?
>
> To the best of our knowledge, this is the first work aiming to formally compare two neural networks jointly. We are happy to make such comparisons should the reviewer provide a reference.
>
> As discussed earlier, in our experiments, we have performed a comparison with performing such analysis independently and shown that our approach is substantially more accurate/tight than looking individually into robustness properties of each network (“Comparison with Independent Analysis” and Figure 2 in Section 4.3).
>
>
> >5. Medical Application Claims It would be good to compare your estimates to some sources of ground truth. Is that feasible?
>
> We have made several experiments on two well-established medical applications, with ground-truth labels provided by medical experts. Our bounds, as mentioned before, are sound and provably correct (formally verified). Therefore, when our framework concludes that  a network consistently makes a correct decision every time the other network does in the entire input region, the conclusion is guaranteed to be correct.
>
> We thank the reviewer for their time. We'd appreciate knowing if our clarifications on contribution have satisfied you and if our clarified perspective might improve your evaluation of our work and your confidence in our work.

---

> > ### Author Response · Authors · 2024-11-30
> >
> > We thank the reviewer again for their effort in reviewing our paper and their time. Given that we are approaching the end of the discussion period, we'd appreciate knowing if our responses have satisfied you and if our clarified perspective might improve your evaluation. Any further questions, comments, or suggestions for enhancing the paper are most welcome.

---

### Meta-Review · Area_Chair_dctA · 2024-12-23

**Metareview:**

The paper introduces a framework for comparing two neural network classifiers with identical input and output domains by quantifying the Relative Output Margin (ROM). ROM measures the consistency and correctness of one network's decisions relative to another over a specified input region. The framework provides provably correct bounds on ROM gains or losses, offering a formal verification method for assessing decision quality across input regions. The authors demonstrate the framework's applicability using datasets such as MNIST, CIFAR-10, and two real-world medical datasets.

The reviewers found the question studied important, the proposed method solid and sound, and the paper well-written. They were concerned about the scalability of the proposed linear programming method, though this is a notoriously hard challenge in formal verification of NNs. Additionally, some reviewers raised questions about the meaning and significance of the LROM measure itself. Despite several rounds of discussion, these concerns were not convincingly addressed. The authors are encouraged to carefully consider these comments to refine the methodology and improve the presentation, ensuring a more convincing and clear argument for future readers.

**Additional Comments On Reviewer Discussion:**

To keep this summary short I will focus on issues that I think are the most crucial, which are about how meaningful the ROM measure is. Reviewer eQgt pointed out that "Similarly, the same ROM can represent drastically different situations. In fact, if the two networks have log OM of +0 and +10, we are comparing a coin toss to an extremely accurate predictor. This is not the same as having two extremely confident networks with log OM of +1000 and +1010, but both situations have log ROM of +10." The authors' response essentially repeats the definition of OM but does not directly address this conceptual question. A few other concerns/questions about the meaningfulness of ROM raised by Reviewers eQgt and tfBm similarly did not get resolved during the discussion phase.

---

> ### Public Comment · ~Anahita_Baninajjar1 · 2025-02-26
>
> We disagree with the statement that our approach equates a “a coin toss” and “an extremely accurate predictor” because 1010-1000=10-0 in terms of logits . Both correspond to extremely accurate predictors if you consider the softmax layer (as is common in softmax based classifiers). Indeed, output margins are defined on the output of the softmax layer, NOT on the logits before the softmax layer. Although it may seem that if the output margins of one pair of networks with logits 0 and 10 differs from another pair with logits of 1000 and 1010, the output margins for both networks are exactly the same (0.5% and 99.5%). Therefore, when comparing the relative output margin after the softmax layer, it does not matter whether the logits are 0-10, 1000-1010 or 1000000-1000010.
>
> That being said, we could as well focus on the logits instead of the softmax outputs.
>
> The fundamental question we try to answer is however “Given two compatible networks and an input region, is it possible to formally prove that one network consistently makes a correct decision every time the other network does, in the entire input region.”
>
> We thank the reviewers and chairs for their efforts. We will strive to better clarify the contributions in the future.

---

### Decision · Program_Chairs · 2025-01-22

Reject